# Ear pinnae in a neotropical katydid (Orthoptera: Tettigoniidae) function as ultrasound guides for bat detection

Christian A Pulver[1†], Emine Celiker[1*†‡], Charlie Woodrow[1], Inga Geipel[2,3,4], Carl D Soulsbury[1], Darron A Cullen[1], Stephen M Rogers[1], Daniel Veitch[1], Fernando Montealegre-Z[1*]

[1]University of Lincoln, School of Life & Environmental Sciences, Joseph Banks Laboratories, Green Lane, Lincoln, United Kingdom; [2]Smithsonian Tropical Research Institute, Balboa, Panama; [3]CoSys Lab, Faculty of Applied Engineering, University of Antwerp, Antwerp, Belgium; [4]Flanders Make Strategic Research Centre, Lommel, Belgium

*For correspondence:
ECeliker@lincoln.ac.uk (EC);
fmontealegrez@lincoln.ac.uk
(FM-Z)

†These authors contributed
equally to this work

Present address: ‡Division
of Mathematics, University
of Dundee, Dundee, United
Kingdom

Competing interest: The authors
declare that no competing
interests exist.

Reviewing Editor: Andrew
J King, University of Oxford,
United Kingdom

**Abstract** Early predator detection is a key component of the predator-prey arms race and has driven the evolution of multiple animal hearing systems. Katydids (Insecta) have sophisticated ears, each consisting of paired tympana on each foreleg that receive sound both externally, through the air, and internally via a narrowing ear canal running through the leg from an acoustic spiracle on the thorax. These ears are pressure-time difference receivers capable of sensitive and accurate directional hearing across a wide frequency range. Many katydid species have cuticular pinnae which form cavities around the outer tympanal surfaces, but their function is unknown. We investigated pinnal function in the katydid *Copiphora gorgonensis* by combining experimental biophysics and numerical modelling using 3D ear geometries. We found that the pinnae in *C. gorgonensis* do not assist in directional hearing for conspecific call frequencies, but instead act as ultrasound detectors. Pinnae induced large sound pressure gains (20–30 dB) that enhanced sound detection at high ultrasonic frequencies (>60 kHz), matching the echolocation range of co-occurring insectivorous gleaning bats. These findings were supported by behavioural and neural audiograms and pinnal cavity resonances from live specimens, and comparisons with the pinnal mechanics of sympatric katydid species, which together suggest that katydid pinnae primarily evolved for the enhanced detection of predatory bats.

## Editor's evaluation

This study combines an impressive combination of experimental and computational approaches to probe the function of the cuticular pinnae, structures that form air-filled cavities around the tympanal ears on the forelegs of bush crickets. In many other species – including mammals – the external ears are known to play a critical role in helping to localize sounds. The results of this study show, however, that the very small resonant cavities formed by the pinnae in one particular bush cricket species are able to boost ultra-high frequency sound waves that lie well above the frequencies used for communicating with conspecifics. This raises the possibility that these structures may have evolved to assist bush crickets to detect the ultrasonic echolocation calls of their bat predators.

## Introduction

Throughout the animal kingdom, the need to localise acoustic cues from predators and prey, as well as signals from conspecifics, is a major selection pressure (*Fay and Popper, 2000*). To determine the location of a sound source, animals with two ears utilize interaural time and amplitude differences. Such binaural auditory systems must satisfy three requirements to function: (1) the distance between the ears must be sufficient to produce recognisable differences in sound arrival time; (2) the ears must be separated by an anatomical structure which is large enough to attenuate sound between them; (3) the ears must be neurologically coupled in order to calculate time and amplitude differences (*Brown, 1984*; *Christensen-Dalsgaard et al., 2021*; *Christensen-Dalsgaard and Manley, 2005*; *Lakes-Harlan and Scherberich, 2015*; *Lauer et al., 2018*; *Suga, 1989*; *Zaslavski, 1999*). However, animals such as insects are too small to exploit diffractive effects of sound on their bodies to perceive minute differences in sound delays and intensities (*Michelsen and Larsen, 2008*). As a result, vastly different species have convergently evolved separate mechanisms of hearing to fulfil similar functions (*Göpfert and Hennig, 2016*; *Köppl et al., 2014*; *Robert, 2005*; *Warren and Nowotny, 2021*), including the detection of ultrasonic frequencies (*Strauß et al., 2014*).

For katydids (or 'bush crickets': Orthoptera: Tettigoniidae), a family with over 8100 species (*Cigliano et al., 2021*), size may be less of a problem as their ears are located in their two forelegs rather than on their body (*Bailey, 1990*), which provides a greater interaural distance and interaural phase difference, meaning that the resulting distance between the ears provides sufficient spatial separation to exceed the wavelengths of incoming conspecific sounds (*Robert, 2005*). Each ear consists of two tympanal membranes on the proximal front tibia (one anterior membrane and one posterior), which are both able to receive sound directly at the external tympanal surface (referred as the external input) but also internally through a long, air-filled tube evolutionarily derived from respiratory trachea known as the acoustic trachea or ear canal (ear canal henceforth; *Figure 1A*). In the internal path, sound enters the ear canal through a specialised opening in the prothorax known as the acoustic spiracle (*Kalmring et al., 2003*). The ear canal's narrowing, exponential horn shape passively amplifies sound pressure (*Celiker et al., 2020a*; *Michelsen et al., 1994*; *Veitch et al., 2021*), reduces propagation sound velocity (*Jonsson et al., 2016*; *Michelsen et al., 1994*; *Veitch et al., 2021*), and leads these decelerated sound waves through the thorax and foreleg to the internal tympanal surface. The combined phase differences of the internal and external paths generate disparities in sound pressure and arrival times on the external and internal surfaces of the tympanal membranes of each ear. Thus, multiple pathways provide the interaural phase differences to reliably encode the angle of the sound source. The katydid ear therefore functions as a pressure – time difference receiver (*Michelsen and Larsen, 2008*; *Robert, 2005*; *Veitch et al., 2021*), unlike the mammalian ear which functions as a single input pressure receiver via the ear canal.

At the external auditory input, many katydid species (>65%, *Cigliano et al., 2021*) possess cuticular pinnae (also referred to as folds, flaps or tympanal covers) partially enclosing one or both of their tympana within an air cavity. Morphologies of cuticular pinnae vary greatly between species (*Figure 1C and D*), but their role(s) remain unclear. Before experimental evidence of the dual input system in katydids was published (*Jonsson et al., 2016*; *Michelsen et al., 1994*), early observations suggested that pinnae aid in determining the direction of sound (*Bailey and Stephen, 1978*; *Autrum, 1963*; *Autrum, 1942*; *Autrum, 1940*). Others suggested that pinnae are merely protective structures sheltering the fragile tympanum (*Pumphrey, 1940*; *Lewis, 1974b*). Subsequently, several authors tested Autrum's hypothesis using electrophysiological techniques and could not demonstrate a role for the pinnae in directional hearing, and instead showed that ear sensitivity depends on sound directed to acoustic spiracles (*Lewis, 1974a*; *Lewis, 1974b*; *Nocke, 1975*; *Eisner and Popov, 1978*; *Hill and Oldfield, 1981*; *Hoffmann and Jatho, 1995*; *Michelsen and Nocke, 1974*; *Shen, 1993*). *Lewis, 1974b* was the first to suggest a role for the pinnae in maintaining a high sensitivity of the organ at high frequencies. Studies of ultrasonic rainforest Pseudophyllinae provided more evidence of principal sound reception for conspecific communication using the external tympanal input instead of their exceptionally small spiracle sizes (*Mason et al., 1991*). It was reported that diffraction of very short wavelengths along the pinnal cavity entrances (or slits, *Figure 1B*) produced the strongest responses when stimuli was presented directly opposite the cavity entrances, and weakest contralaterally to the same stimuli. This difference in intensity between the two ears potentially contributes to directional orientation in rainforest katydids.

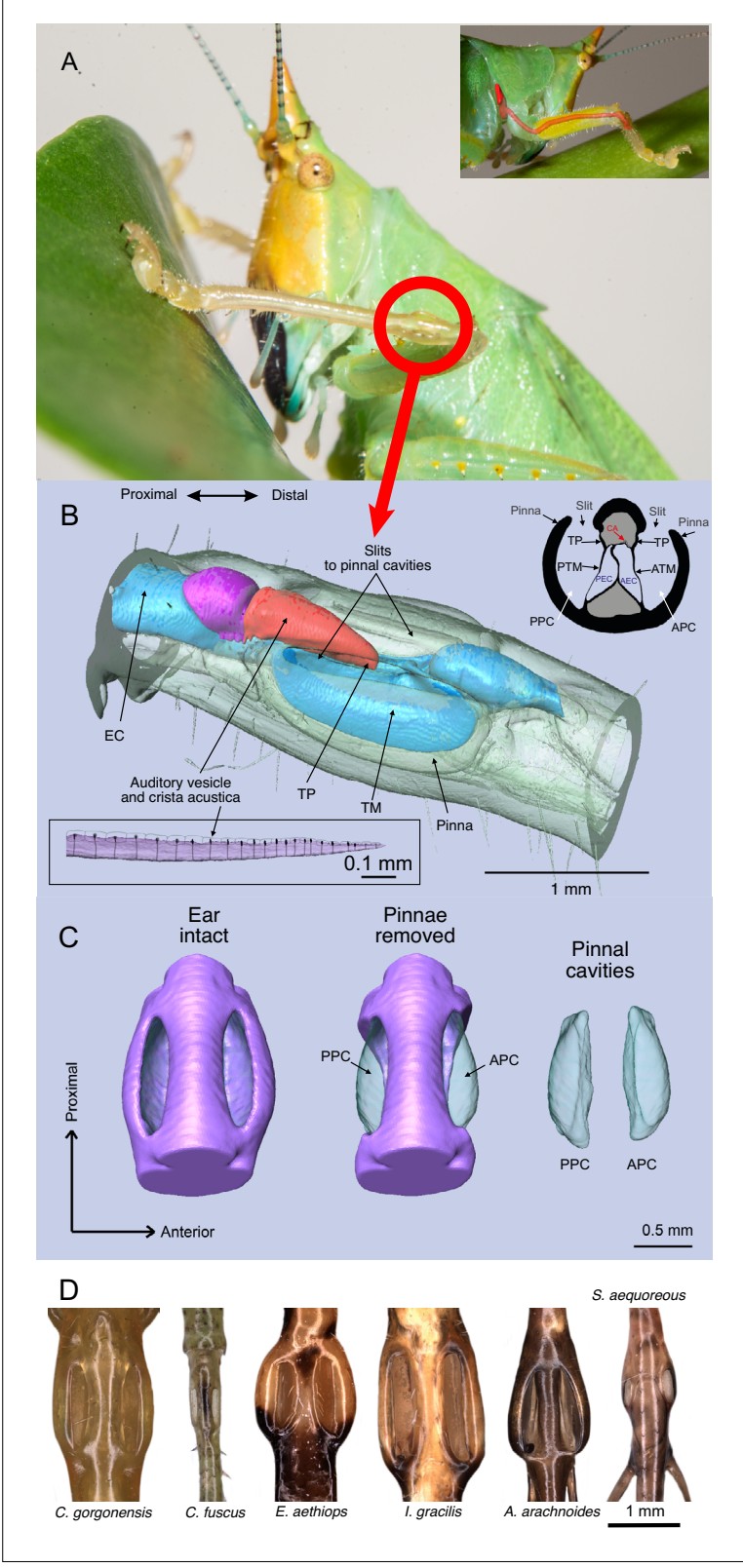

**Figure 1.** The ear of *Copiphora gorgonensis*. (**A**) Location of the ear in the foreleg with a smaller panel illustrating the ear canal extending from the prothorax (acoustic spiracle) to the femoro-tibial "knee" joint through the foreleg. (**B**) 3D reconstruction of the tympanal organ of *C. gorgonensis*, showing external and internal structures. Inset in the upper right corner shows a cross section through the ear, inset in the lower left corner shows a lateral view of

*Figure 1 continued on next page*

*Figure 1 continued*

the *crista acustica* with sensory cells. EC = ear canal, TP = tympanal plate, ATM = anterior tympanal membrane, PTM posterior tympanal membrane, PEC = posterior ear canal division AEC = anterior ear canal division, CA = crista Acustica, AV = auditory vesicle. APC = anterior pinnal cavity, PPC = posterior pinnal cavity; (**C**) 3D anatomy of the ear, with pinnae present, removed, and pinnal cavity volumes; (**D**) Examples of cuticular pinnae of various katydids from three ensiferan subfamilies: (**L–R**): *C. gorgonensis* (Conocephalinae), *Conocephalus fuscus* (Conocephalinae), *Eubliastes aethiops* (Pseudophyllinae), *Ischnomela gracilis* (Pseudophyllinae), *Arachnoscelis arachnoides* (Meconematinae), and *Supersonus aequoreus* (Meconematinae).

Here, we investigate the role of cuticular pinnae using the neotropical katydid *Copiphora gorgonensis* (*Montealegre-Z and Postles, 2010*). Males of this species produce a pure-tone song at 23 kHz to attract females. We integrated experimental biophysical measurements based on micro–scanning laser Doppler vibrometry (LDV) and micro-computed tomography to simulate the function of the cuticular pinnae and how they contribute to auditory orientation in this katydid. These approaches were applied to 3D printed models of the ear, and scaled experiments were performed to validate the simulations. We investigated if: (1) the direction of incidence of the sound stimulus is a function of the sound wave directly accessing the tympana through the cavity entrances; (2) the pinnal cavities produce sound pressure gains that act externally on the tympana; (3) tuning properties of the pinnal cavities are a result of pinnal geometry and can be predicted by the volume and/or entrance size of the cavity; (4) neural and behavioural responses to resonant frequencies of the cavities substantiate experimental and numerical conclusions; (5) calls from co-occurring, predatory bats match the resonant frequencies of the pinnal cavities.

We hypothesized that tympanal pinnae function as detectors for high ultrasonic frequencies. The small cavities formed by the pinnae act as Helmholtz-like resonators able to capture and amplify diminishing ultra-high frequency sound waves.

## Results

### The effect of pinnae on temporal dynamics of sound arrival at the tympana

We investigated the role of pinnae in sound capture by testing how the incidence direction of the sound stimulus induced tympanal displacement at three frequencies (23, 40, and 60 kHz) with the cuticular pinnae intact and later ablated. Frequencies above 60 kHz were not tested due to the acoustic limitations of the experimental setup (see Materials and methods). A total of 2736 measurements were performed on 13 ears (1512 measurements for four male specimens; 1224 for three female specimens).

We found a significant interaction between the presence of pinnae with angle of incidence (21° semicircle azimuth frontal to ear; *Figure 2A*) and with frequency (*Table 1*). Post-hoc analysis showed that pinnae significantly delayed the time of arrival at 23 kHz from 0.56 ± 0.05 µs to 0.55 ± 0.05 µs (*t*-ratio = –11.15, *p* < 0.001), and at 40 kHz from 0.56 ± 0.05 µs to 0.56 ± 0.05 (*t*-ratio = –7.43, *p* < 0.001), but not at 60 kHz (*t*-ratio = –1.86, *p* = 0.063). Thus, the effect of pinnae on arrival times was less pronounced at increasing frequencies. Sound arrived at the posterior tympanum ~2 µs later than at the anterior tympanum, a significant delay, and mean displacement amplitude at the posterior tympanum was also significantly lower (by 21.5%; *Table 1*).

For displacement amplitude, there was a significant interaction between the presence of pinnae and frequency (*Table 1*). Post-hoc analysis showed maximum displacement amplitudes at 23 kHz with both intact and ablated pinnae, but the greatest displacement with the pinnae ablated (*t*-ratio = 3.20, *p* < 0.001; *Figure 2B*). This demonstrates that pinnae do not enhance auditory perception of the carrier frequency in *C. gorgonensis*, and that the observed displacement, even after ablation, results from the fact that tympanal natural resonance produces maximum vibrational amplitude at 23 kHz, the carrier frequency of the species call as demonstrated by *Jonsson et al., 2016*; *Montealegre-Z et al., 2012*. There were no differences in displacement at either 40 kHz (*t*-ratio = 0.84, *p* = 0.399; *Figure 2B*) or 60 kHz (*t*-ratio = –0.61, *p* = 0.540; *Figure 2B*) regardless of the presence or absence of pinnae. We also found a significant interaction between the presence of pinnae and angle of incidence, with pinnae increasing arrival time with increased angle (*Table 1*). Responses were strongest for sound

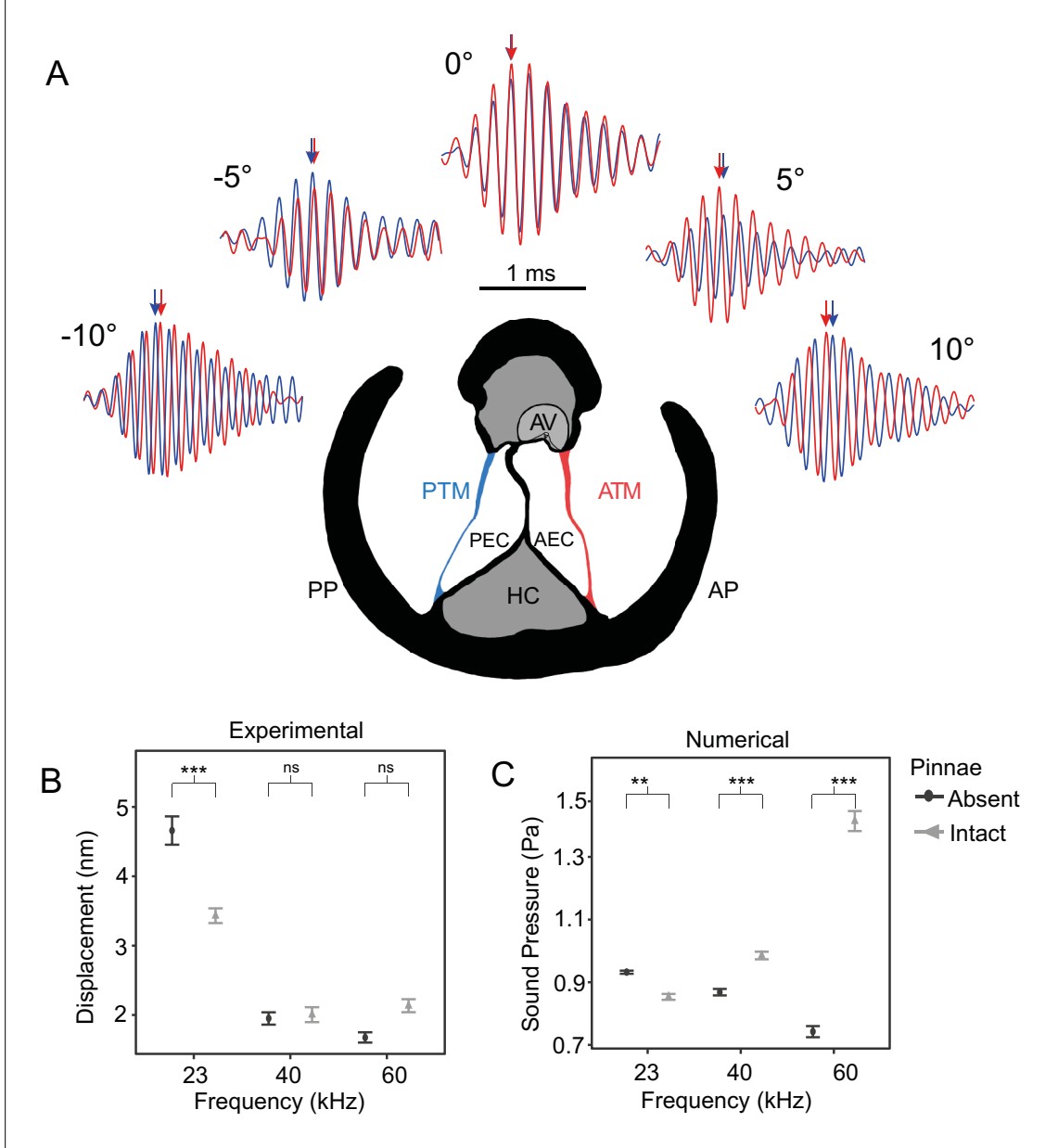

**Figure 2.** The effect of pinnae in the time domain and numerical simulations. (**A**) Time plots from five incidence angles for the 60 kHz test sound illustrating changes in oscillation phase between the anterior (ATM, in red) and posterior (PTM, in blue) tympana of the same ear. Notice the phase difference of 0.25 cycles is 90° at −10° and 10°. An anatomical cross section of the ear is shown with each tympanum (ATM and PTM), auditory vesicle (AV), posterior and anterior bifurcated ear canal branches (PEC and AEC), haemolymph channel (HC) and posterior and anterior pinnal structures (PP and AP). (**B**) Mean displacement amplitudes (nm) of the tympanal membranes for each tested frequency (23, 40, and 60 kHz) with and without the presence of cuticular pinnae (n = 9 ears). (**C**) Cavity-induced pressure gains with pinnae compared to sound pressure (Pa) predictions with the pinnae ablated from numerical models using Comsol Multiphysics (17 ears; 10 females, 7 males). For means comparison plots (**B**) & (**C**), significance symbols from post hoc analyses: '***' 0.001, '**' 0.01, '*' 0.05, 'ns' 0.1, and ' ' 1. Grey bars with cuticular pinnae and black bars without cuticular pinnae showing standard error.

The online version of this article includes the following figure supplement(s) for figure 2:

**Figure supplement 1.** Diagram of experimental arena with a mounted *Copiphora gorgonensis* (not drawn to scale) on specialised (input isolating) platform with a rotating probe-tipped loudspeaker perpendicular to the tympanal septum and single point sensor heads (OFV-534) in perpendicular position.

**Figure supplement 2.** Illustration of experimental arena with a cross section of the copiphorine ear positioned in relation to the single point laser sensor heads with magnifying lenses (SL1 and SL2) and the rotating probe-loudspeaker.

**Table 1.** Linear mixed models (LMM) of experimental and numerical simulation data.
Parameters showing effects of angle, pinnae, frequency, tympanum, angle × pinnae and pinnae × frequency for time domain data (experimental time and displacement) and sound pressure (numerical simulations and 3D print models). Experimental models $n$ = 13 ears. Numerical model $n$ = 17 ears. 3D model $n$ = 4 ears.

| Model | Parameter | F | p |
|---|---|---|---|
| Experimental Time Domain | Angle (polynomial) | 5.35 | 0.005 |
| | Pinnae (Y/N) | 254.60 | <0.001 |
| | Frequency | 2097.26 | <0.001 |
| | Tympanum | 5.07 | 0.024 |
| | Angle × Pinnae | 4.47 | 0.012 |
| | Pinnae × Frequency | 31.93 | <0.001 |
| Experimental Displacement | Angle (polynomial) | 3.29 | 0.037 |
| | Pinnae (Y/N) | 0.90 | 0.344 |
| | Frequency | 270.57 | <0.001 |
| | Tympanum | 17.32 | <0.001 |
| | Angle × Pinnae | 4.89 | 0.008 |
| | Pinnae × Frequency | 5.41 | 0.004 |
| Numerical Sound Pressure | Angle (polynomial) | 0.72 | 0.489 |
| | Pinnae (Y/N) | 336.55 | <0.001 |
| | Frequency | 69.29 | <0.001 |
| | Tympanum | 0.02 | 0.879 |
| | Angle × Pinnae | 1.31 | 0.271 |
| | Pinnae × Frequency | 761.46 | <0.001 |
| 3D Model Sound Pressure | Pinnae (Y/N) | 1175.9 | <0.001 |
| | Frequency | 314.58 | <0.001 |
| | Tympanum | 0.01 | 0.9111 |
| | Pinnae × Frequency | 296.70 | <0.001 |

presented perpendicular to each respective cavity ($n$ = 7; average 3.08 ± 2.91 nm at 10°, average 2.90 ± 2.98 nm at –9°) with the lowest displacement amplitudes occurring when sound was directed at the region of the dorsal cuticle between the cavities, also referred as 'point zero' ($n$ = 7; average 1.99 ± 1.90 nm at –1°) with the pinnae intact due to cuticle obstructing the response of the tympanal membrane. In contrast, point zero and adjacent angles showed the greatest displacement amplitude with the pinnae ablated ($n$ = 7; average 3.04 ± 3.42 nm at –1°) with incident angles on either side of point zero returning a gradually decreasing response to the stimulus ($n$ = 7; average 2.73 ± 3.27 nm at 10°, average 2.54 ± 2.57 nm at –10°).

Phase angle ($\varphi°$) was calculated from the absolute value of the difference between the vibrations of the anterior and posterior tympana per recording ($n$ = 7; 1532 in total). Pinnae maintained mean $\Delta \varphi°$ at 80.9° for 23 kHz, 88.8° for 40 kHz, and 84.1° for 60 kHz, but with the pinnae ablated, phase differences were smaller particularly at 60 kHz ($\Delta \varphi°$ at 62.7° for 23 kHz, 78.7° for 40 kHz, and 49° for 60 kHz).

**Table 2.** Measured parameters of the ear of *C. gorgonensis* (*n* = 8 ears; 3 females, 2 males).
Given are mean values (± SD). Abbreviations: APC = anterior pinnal cavity; PPC = posterior pinnal cavity.

| APC volume (mm³) | PPC volume (mm³) | Distance between slits (mm) | Cross-sectional width of foreleg below ear (mm) | Cross-sectional width of ear (mm) | APC slit area (mm²) | PPC slit area (mm²) | Protrusion of anterior pinna (mm) | Protrusion of posterior pinna (mm) |
|---|---|---|---|---|---|---|---|---|
| 0.14 (±0.01) | 0.15 (±0.01) | 0.42 (±0.03) | 0.84 (±0.02) | 1.14 (±0.35) | 0.16 (±0.01) | 0.16 (±0.01) | 0.39 (±0.02) | 0.45 (±0.03) |

## Evidence of cavity-induced sound pressure gains

### Anatomical measurements of the external tympanal input

The anatomical features of the ear were measured to predict resonance and compare intraspecific variation in pinna size (*Table 2*). 2D measurements of the area of the pinnal entrance (slit), distance between the centre of the ear (septum) and edge of the pinna (pinnal protrusion), and distance between slits (septum width) were studied using an Alicona Infinite Focus microscope. 3D measurements of the cavities and cross section of the ears were performed with the micro-computed tomography scanner using the software Amira-Aviso 6.7 (*n* = 8 ears; 3 females and 2 males). We found that the average size of the slits (0.16 ± 0.01 mm²) and cavities (0.14 ± 0.01 mm³) were nearly identical between the anterior and posterior pinnae. The posterior pinnae (0.44 ± 0.03 mm) was wider than the anterior pinnae (0.39 ± 0.02 mm). The mean cross-sectional width of the ear was 1.14 ± 0.35 mm.

### Pinnal cavity resonance calculations

We used slit area and cavity volume to estimate the resonance of the pinnal cavities (*Table 2*). This was calculated with the assumption that the 2D slit entrances were a perfect circle (to determine radius) and the 3D cavity acted as a cylindrical tube using a neckless Helmholtz resonance equation. Here, *c* is speed of sound in air (343 m s⁻¹), cross-sectional area of the entrance with radius *r*, 1.85 is the correction length of the neck and *V* denotes the volume of the resonator/cavity (*Rossing and Fletcher, 2004*).

$$f(h) = \frac{c}{2\pi}\sqrt{\frac{1.85r}{V}}$$

The pinnal cavities (*n* = 8) showed a neckless Helmholtz resonance of 94.28 ± 3.53 kHz for the anterior cavity and 91.69 ± 3.93 kHz for the posterior cavity. These calculations suggest that the pinnal cavities resonate far closer to bat hunting frequencies than to the 23 kHz calling song frequency of *C. gorgonensis*.

### 3D printed model time and frequency domain measurements of pinnal cavities

3D printed scaled models of the ear were used to measure sound pressure gains and resonances, to overcome the limitations imposed by the small size of the animals (*Figure 3*). 3D printed ears (*n* = 8; 4 prints from males and 4 prints from females, 2 ears each, ± pinnae) were printed at a scale of 1:~11.5 and the acoustic stimuli were scaled by the same factor for pure tones (2.01 kHz for 23 kHz, 3.50 kHz for 40 kHz, 5.25 kHz for 60 kHz, and 9.63 kHz for 110 kHz) and for broadband (2–15 kHz for 11.5–170 kHz). Sound pressure (dB) did not significantly differ between the anterior and posterior pinnal cavities, but it was significantly affected by the interaction between frequency and the presence/absence of pinnae (*Table 1*). Pinnae increased sound pressure across all frequencies tested, and this effect was greatest at higher frequencies (23 kHz: *t*-ratio = –2.54, *p* = 0.014; 40 kHz: *t*-ratio = –8.69, *p* < 0.001; 60 kHz *t*-ratio = –15.66, *p* < 0.001; 110 kHz *t*-ratio = 41.70, *p* < 0.001; *Figure 4A*; *Video 1*). Overall, the greatest pressure gains were detected at 101.47 ± 3.43 kHz for both the anterior (26.33 ± 4.06 dB) and posterior pinnal cavities (30.04 ± 1.34 dB) with the pinnae intact. With the pinnae ablated, the greatest pressure gain was at 101.41 ± 0.86 kHz for both the anterior (9.69 ± 0.87 dB) and posterior (9.83 ± 0.97 dB) pinnal cavities. Stimulation of the 3D printed models with broadband sound showed that both pinnal cavities resonate across a broad range of high ultrasonic frequencies between around 60 and 120 kHz. When the pinnae were removed the sound pressure gain was significantly reduced, although a low amplitude resonance persists due to a recessed V-shaped cavity that

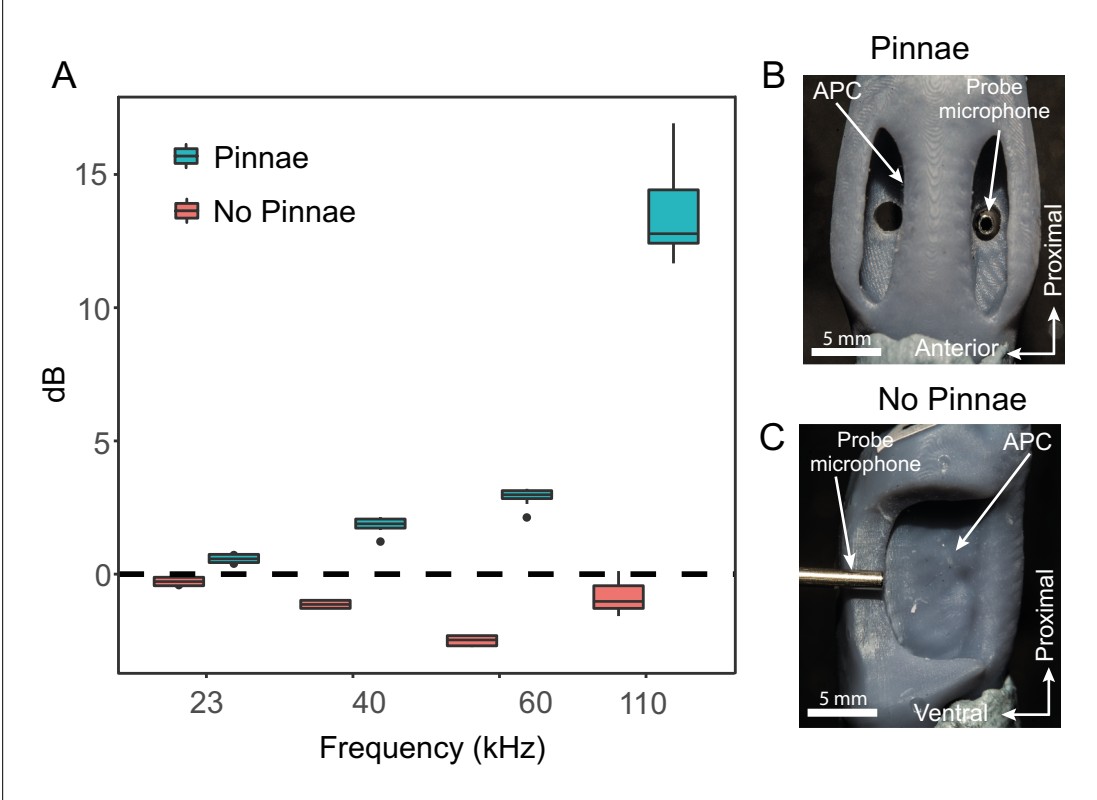

**Figure 3.** Acoustic experiments with 3D printed scaled ear models. (**A**) Sound pressure gains (dB SPL) of 3D printed ears (*n* = 8; 4 males and 4 females) calculated from scaled time domain recordings for 23, 40, 60, and 110 kHz 4-cycle pure tones. (**B**) An example of a 3D printed ear model with pinnae present (dorsal view) showing the probe microphone inside the posterior tympanum. (**C**) An example of a 3D printed ear model with pinnae ablated (anterior lateral view), showing probe microphone placement.

The online version of this article includes the following figure supplement(s) for figure 3:

**Figure supplement 1.** Pinnal cavity relative gain (dB) of sympatric katydid species with auditory pinnae, from the island of Gorgona, Colombia, in response to scaled broadband chirps.

remains after pinnal ablation. See next section, *Figure 4E and F* and *Videos 2 and 3* for more details, and for a comparison with the numerical simulations.

## Tuning properties of the pinnal cavities
### Numerical modelling

Using life-scale 3D geometries of each experimental ear (*n* = 17 ears; 8 with pinnae, 9 with pinnae ablated), we used Finite Element Analysis (FEA) to simulate sound pressure gains and the effect of incident angle at frequencies exceeding those experimentally possible with live specimens (see Materials and methods). For sound pressure measurements there was a significant interaction between the presence of pinnae and frequency (*Table 1*). At 23 kHz, ears without pinnae received significantly higher sound pressures (t-ratio = 3.45, *p* < 0.001), but the effect was reversed at 40 kHz (t-ratio = –5.94, *p* < 0.001) and 60 kHz (t-ratio = –28.52, *p* < 0.001), with differences increasing as frequency increased (*Figure 2C*). Sound pressure level was not significantly affected by the angle of sound incidence (–10°, –5°, 0°, 5°, 10°), and did not significantly differ between the anterior and posterior pinnal cavities (*Table 1*).

Simulated sound pressure gains and their distribution maps (*Figure 4A and B*) showed the greatest sound pressure gain at a mean value of 118 kHz (anterior pinnal cavity 121 kHz, posterior pinnal cavity 115 kHz), and these gains were reduced or lost entirely when the pinnae were removed (*Figure 4D*; *Table 1*). These simulations validate our experiments with 3D-printed models with scaled sound frequencies (*Figure 4E and F*).

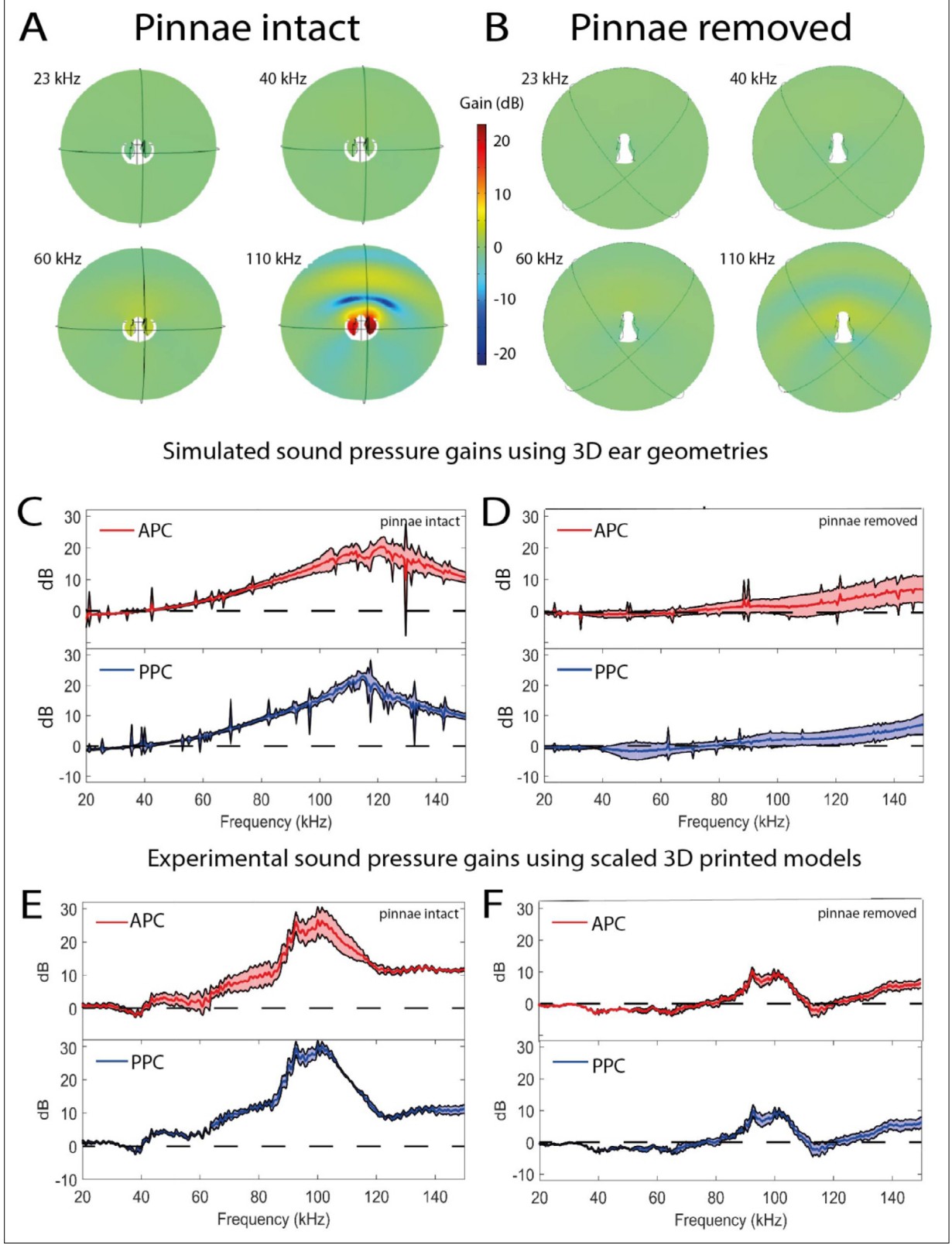

**Figure 4.** Sound pressure gains measured by numerical simulations of sound capture in the pinnal cavities (using Comsol Multiphysics) and experimentally using printed 3D-scaled ear geometries. Panels (**A**), (**C**), and (**E**) depict cavity-induced sound pressure distribution and gains with pinnae, panels (**B**), (**D**), and (**F**) represent sound pressure gains without the pinnae. (**A and B**) Numerical simulations obtained on 3D ear geometries. Cross-section of the ear of *Copiphora gorgonensis* with the pinnae intact (**A**) and ablated (**B**). Sound pressure intensities depicted with colours for

*Figure 4 continued on next page*

*Figure 4 continued*
simulations of 23, 40, 60, and 110 kHz. Low sound pressure dB (blue) to high sound pressure dB (red) distributions inside and outside the cavities. (**C and D**) Simulated sound pressure gains (dB SPL) in the frequency ranges of 20–150 kHz for each tympanum. (**E and F**) Relative dB gain of the pinnal cavities in the 3D printed ears. APC in red and PPCin blue.

The online version of this article includes the following figure supplement(s) for figure 4:

**Figure supplement 1.** Comsol Multiphysics reconstruction of ear geometry.

The effects of angle, pinnae, tympanum, interaction of angle and pinnae, and the interaction of pinnae and frequency were not significant on arrival times. However, the effect of frequency was significant on arrival times: We found longer arrival times at 23 kHz (0.068 ± 0.018 ms) with decreasing arrival times at increasing frequencies at 40 kHz (0.039 ± 0.008 ms), at 60 kHz (0.026 ± 0.005 ms). 23 vs 60 kHz *t*-ratio = 30.739, *p* < 0.001; 23 vs 60 kHz: *t*-ratio = 45.857, *p* < 0.001; 40 vs 60 kHz: *t*-ratio = 15.117, *p* < 0.001.

## Tympanal response to broadband stimulation
For broad tympanal responses, we exposed seven specimens with intact pinnae to broadband periodic chirp stimulation in the range 20–120 kHz in a free sound field and recorded the vibrations of all four tympana across both ears using a micro-scanning laser Doppler vibrometer. There was a relatively stable response (measured as velocity per sound pressure) of the tympanal membranes between 20 and 70 kHz. However, above 80 kHz the tympanal response increased dramatically with resonant peaks at 107.84 ± 3.74 kHz for the posterior tympanum and 111.13 ± 4.24 kHz for the anterior tympanum (*Figure 5A*). However, the gain of the posterior tympanum was about three-fold larger than that of the anterior tympanum.

## Behavioural and neural responses to broadband stimulation
### Behavioural audiograms
Behavioural audiograms of startle behaviour were obtained from nine tethered females walking on a treadmill. Audiograms were obtained with stimuli in the range 20–120 kHz. Audiograms showed that the startle response of females decline sharply for stimuli between 20 kHz and 35 kHz, however, response increases at around 35 kHz, and remains essentially constant at higher frequencies over the entire tested frequency range (*Figure 5B*; *Table 3*). A decline in threshold was found at the resonances of the pinnal cavities (90 kHz to 120 kHz) 59.28 ± 1.80 dB SPL (*Figure 5B*).

### Neural audiograms
Extracellular whole auditory nerve recordings, made with suction electrodes, were used to

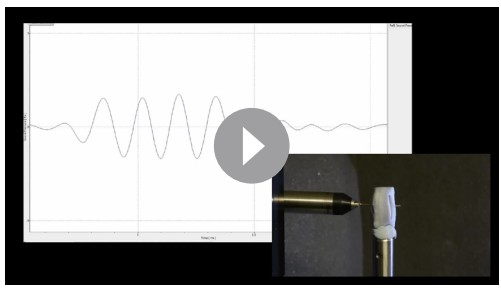

**Video 1.** 3D print ear with microphone. Video recording of probe microphone placement inside the 3D printed ear of *C. gorgonensis*. A digital micromanipulator with a holder restraining the 3D printed ear moved the ear along the probe tip. The microphone remained stationary. Scaled stimuli 6.67 kHz (60 kHz).
https://elifesciences.org/articles/77628/figures#video1

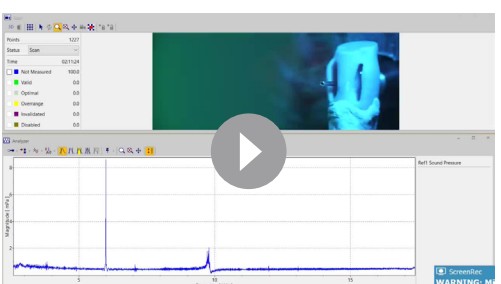

**Video 2.** 3D print ear with microphone receiving broadband chirp. 3D printed ear of *C. gorgonensis* (1:11.512) receiving a scaled broadband chirp of 2.6–17 kHz (corresponding to 30–200 kHz) as the ear is moved into position with the probe microphone inside the cavity. Gain shown in magnitude (mPa). (Note: printed ear and broadband chirp frequency range shown in video are not representative of actual experiments).
https://elifesciences.org/articles/77628/figures#video2

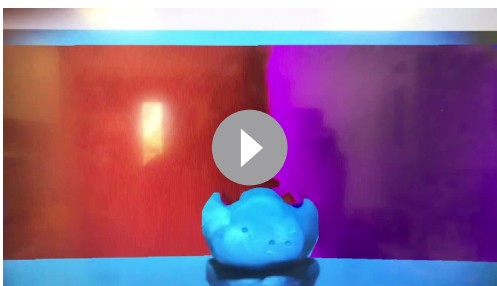

**Video 3.** 3D print ear refractometry. Quantitative imaging of acoustic waves using refracto-vibrometry in the field around the 3D printed ear of *C. gorgonensis* (*Malkin et al., 2014*). Screen recording software of scaled stimuli 9.63 kHz (110 kHz). Note the wave passing over the ear and the piston motion of the air inside showing the effect of the Helmholtz resonator. https://elifesciences.org/articles/77628/figures#video3

produce neural audiograms (*Figure 5C*). The auditory nerve is a mixed nerve, containing the axons of many neurons beside those of auditory afferents, leading to high levels of activity unrelated to auditory stimuli. Furthermore, the high firing rates and small amplitudes of auditory afferent action potentials spread across a population of responsive afferents meant that individual action potentials could not be resolved (*Figure 5—figure supplement 1A*). Instead, the sum neuronal activity in the auditory nerve during sound stimuli was compared with that during silent intervals. Responsiveness was measured by root-mean-square transforming the data (time constant = 0.66 ms) and measuring the area under the curve (*Figure 5—figure supplement 1A*; red, during sound stimulation, blue in between sound stimuli). Auditory stimulation produced significantly greater responses in the neural audiogram recordings compared to neuronal activity during silent periods (*Figure 5—figure supplement 1B*, coloured mesh and grey mesh, respectively) for most combinations of sound frequency and intensity (*Figure 5—figure supplement 1B*, white symbols). Only a small number of the stimuli failed to produce a significant difference in neuronal response, which occurred when frequency was high and sound pressure low (*Figure 5—figure supplement 1B*, black symbols).

At every SPL, the largest responses were seen at the calling song frequency of 23 kHz (*Figure 5—figure supplement 1B*). Taking 70 dB as a representative SPL (*Figure 5C*), the response was 62.7 ± 15.8 µVs during stimulation, which was 86.1% higher than the equivalent off response (*Figure 5—figure supplement 1B*). There was generally a gradual falling away of responsiveness as stimulus frequency increased above 23 kHz: the response to 40 kHz stimulation was 51.8 ± 12.2 µVs; at 60 kHz it was 49.1 ± 14.1 µVs and at 80 kHz stimulation 44.5 ± 12.3 µVs, but measured responses to sound were still substantially above background activity. At 100 kHz, a frequency used by co-occurring echolocating bats, the response of 47.5 ± 12.7 µVs was 43% greater than background activity (and responses at 100 kHz were still resolvable against background activity even for the quietest sound pressure of 46 dB; *Figure 5—figure supplement 1B*). The weakest set of responses was to 120 kHz, which were not distinguishable from the background until above 70 dB SPL (*Figure 5—figure supplement 1B*), but nevertheless demonstrated that very high ultrasonic frequencies can be detected in *C. gorgonensis* if sufficiently loud.

## Echolocation calling frequencies of co-occurring bats

We compared the ultrasonic hearing range of *C. gorgonensis* to the echolocation frequencies of the most common co-occuring insectivorous gleaning bats (*Murillo et al., 2014*), which were recorded in a previous study (*Geipel et al., 2021*). *Gardnerycteris crenulatum* emits multi-harmonic, frequency-modulated (FM) echolocation calls with a call duration of 0.69 ± 0.2 ms, a peak frequency (frequency with maximum amplitude) of 71.1 ± 4.1 kHz and minimum and maximum frequencies (lowest frequency below and highest frequency above the peak frequency with a threshold of –20 dB) of 63.2 ± 3.4 kHz and 95.9 ± 4.8 kHz, respectively (*n* = 50 calls, 1 individual; *Geipel et al., 2021*). *Tonatia saurophila* produces multi-harmonic FM-calls with a duration of 0.69 ± 0.16 ms, a peak frequency at 71.1 ± 8.9 kHz and minimum and maximum frequencies at 34.9 ± 10.4 and 99.2 ± 9.4 kHz, respectively (*n* = 50 calls, 1 individual; *Geipel et al., 2021*). The multi-harmonic FM-calls of *M. microtis* (previously known as *M. megalotis*) have a duration of 0.57 ± 0.04 ms, with a peak frequency at 97.6 ± 5.0 kHz and minimum and maximum frequencies at 60.3 ± 1.8 and 136.4 ± 5.0 kHz, respectively (*n* = 350

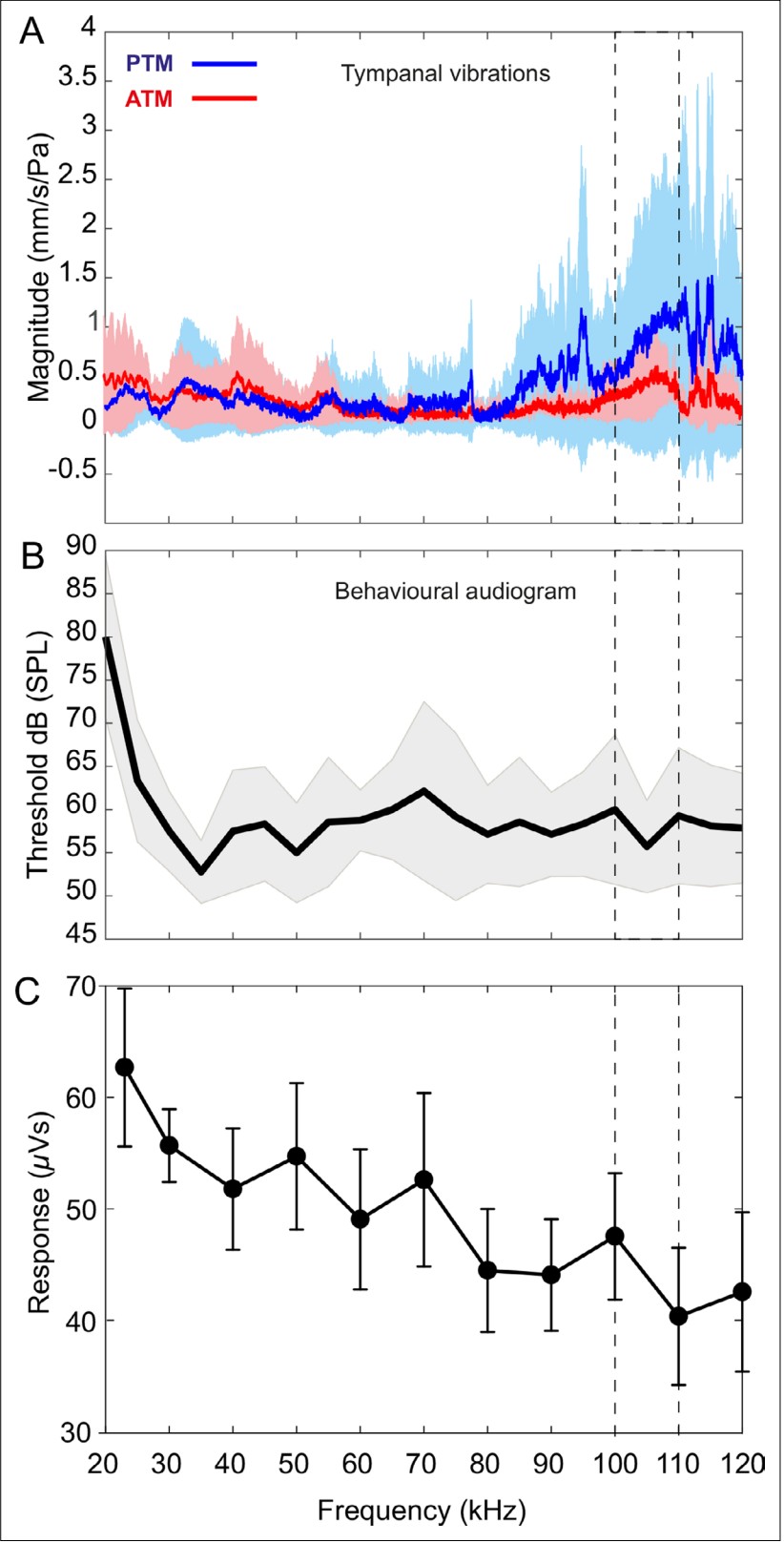

**Figure 5.** Tympanal tuning, behavioural and neural audiograms of *Copiphora gorgonensis*. (**A**) Vibrational responses to broadband chirps (20–120 kHz) of real tympanal membranes (*n* = 7; 14 ears; four males and three females) of live *C. gorgonensis*. Maxima resonance peaks at 107.84 ± 3.74 kHz for the posterior tympanum and 111.13 ± 4.24 kHz for the anterior tympanum. Blue bar for PTM and red bar for ATM. (**B**) Black outline with

*Figure 5 continued on next page*

*Figure 5 continued*

grey shadow indicate the behavioural audiogram of ultrasound response in nine (*n* = 9) female *C. gorgonensis*. Note the drop in threshold within the pinnal frequency range (within the dotted lines) which indicates increased sensitivity. Black outline shows mean vector of SPL response at a particular frequency, shaded area represents the standard deviation across measured SPL. (**C**) Mean ± SEM neural responses at 70 dB across all sound frequencies tested (*n* = 5). Dotted lines indicate high-frequency sensitivity in each measurement, within the range of pinnal resonances.

The online version of this article includes the following figure supplement(s) for figure 5:

**Figure supplement 1.** Neural audiogram showing electrical activity in extracellular recordings of the auditory nerve during sound stimulation as sound pressure intensity (dB SPL) and sound frequency (kHz) are systematically altered.

calls, 7 individuals; *Geipel et al., 2021*). Single calls of each species are presented in *Figure 6* and *Figure 6—figure supplement 1B*.

## Discussion

Tympanal pinnae are present across katydid species, but their function has previously remained unclear. We have shown that, in the model species *Copiphora gorgonensis*, pinnae increase the gain of high ultrasonic frequencies, likely for enhanced detection of their echolocating bat predators. Pinnae serve to expand the auditory dynamic range of the katydid ear beyond the lower frequencies enhanced by the ear canal, enabling the same auditory organ to detect both conspecifics and predators with calling/hunting frequencies nearly an order of magnitude apart. Although these findings are based on a single species, *C. gorgonensis*, which uses low pure tone ultrasonic signals, we cannot reject the possibility that other pinnae-bearing species with broadband frequency calling songs might use the ultrasonic component of their calls for directional hearing using the external sound ports. If high-frequency cues in such katydids provide directional information required for phonotaxis, pinnae could also shed light into the directional mechanism used to detect bats.

In all our experiments, the presence of pinnae had a significant effect on reception of ultrasonic signals above 60 kHz. Further, the extent of the pinnal contribution to tympanal displacement amplitude depended on the incident angle of the sound source at frequencies ≤60 kHz, with pinnae delaying arrival times at the maximum indirect angles (–10° and 10°, *Figure 2A*). The strong differences in the experimental and numerical analysis shown in *Figure 2B, C* happened because the data shows mechanical responses of the tympanum (*Figure 2B*) while the numerical data predicted sound

**Table 3.** Raw data for the behavioural audiogram of ultrasound response in nine female *C. gorgonensis*. NaN denotes that no response was shown to a particular stimulus. Mean and standard deviation calculated ignoring missing data (NaN) for each frequency in the lower rows of the table. All values in dB SPL.

| ID | Frequency (kHz) | | | | | | | | | | | | | | | | | | | | |
|---|---|---|---|---|---|---|---|---|---|---|---|---|---|---|---|---|---|---|---|---|---|
| | 20 | 25 | 30 | 35 | 40 | 45 | 50 | 55 | 60 | 65 | 70 | 75 | 80 | 85 | 90 | 95 | 100 | 105 | 110 | 115 | 120 |
| F1 | 90 | 55 | NaN | 50 | NaN | 55 | 50 | 50 | NaN | NaN | NaN | NaN | 55 | NaN | 55 | NaN | 50 | 50 | 55 | 50 | 60 |
| F2 | 85 | 60 | 55 | 55 | 55 | 50 | 55 | 55 | 55 | 55 | 55 | NaN | 60 | 55 | 60 | 55 | 55 | 55 | 55 | 55 | 55 |
| F3 | 80 | 60 | 60 | 60 | 60 | 60 | 60 | 60 | 60 | 60 | 60 | 60 | 60 | 60 | NaN | 60 | NaN | 60 | 60 | 55 | 60 |
| F4 | 75 | 65 | 65 | 55 | 65 | 65 | 65 | 65 | 65 | 65 | 65 | 65 | 65 | 65 | 65 | 65 | 70 | NaN | 60 | 65 | NaN |
| F5 | 60 | 70 | 50 | 50 | 70 | 70 | 50 | 70 | 60 | 70 | 70 | 55 | 50 | 70 | NaN | NaN | NaN | 55 | 75 | 50 | 70 |
| F6 | 75 | 75 | 55 | 50 | 50 | 60 | NaN | NaN | 60 | 55 | 80 | 75 | NaN | 50 | 60 | NaN | 60 | NaN | 60 | 70 | 50 |
| F7 | 80 | 70 | 60 | 50 | 55 | 55 | 50 | 60 | 55 | 55 | 50 | 50 | 50 | 50 | 50 | 65 | 65 | 55 | NaN | 60 | 55 |
| F8 | 85 | 55 | 55 | 55 | 50 | 50 | NaN | NaN | 55 | 60 | NaN | 50 | NaN | NaN | 55 | 50 | 70 | 65 | 50 | NaN | NaN |
| F9 | 90 | 60 | 60 | 50 | 55 | 60 | 55 | 50 | 60 | NaN | 55 | NaN | 60 | 60 | 55 | 55 | 50 | 50 | NaN | 60 | 55 |
| Mean | 80 | 63 | 57.5 | 53 | 57.5 | 58 | 55 | 59 | 58.8 | 60 | 62.1 | 59 | 57 | 58.6 | 57.1 | 58.3 | 60 | 56 | 59 | 58 | 58 |
| STD | 9 | 7 | 4.63 | 4 | 7.07 | 6.6 | 5.77 | 7.5 | 3.54 | 5.77 | 10.4 | 9.7 | 5.7 | 7.48 | 4.88 | 6.06 | 8.7 | 5.3 | 7.9 | 7 | 6.4 |

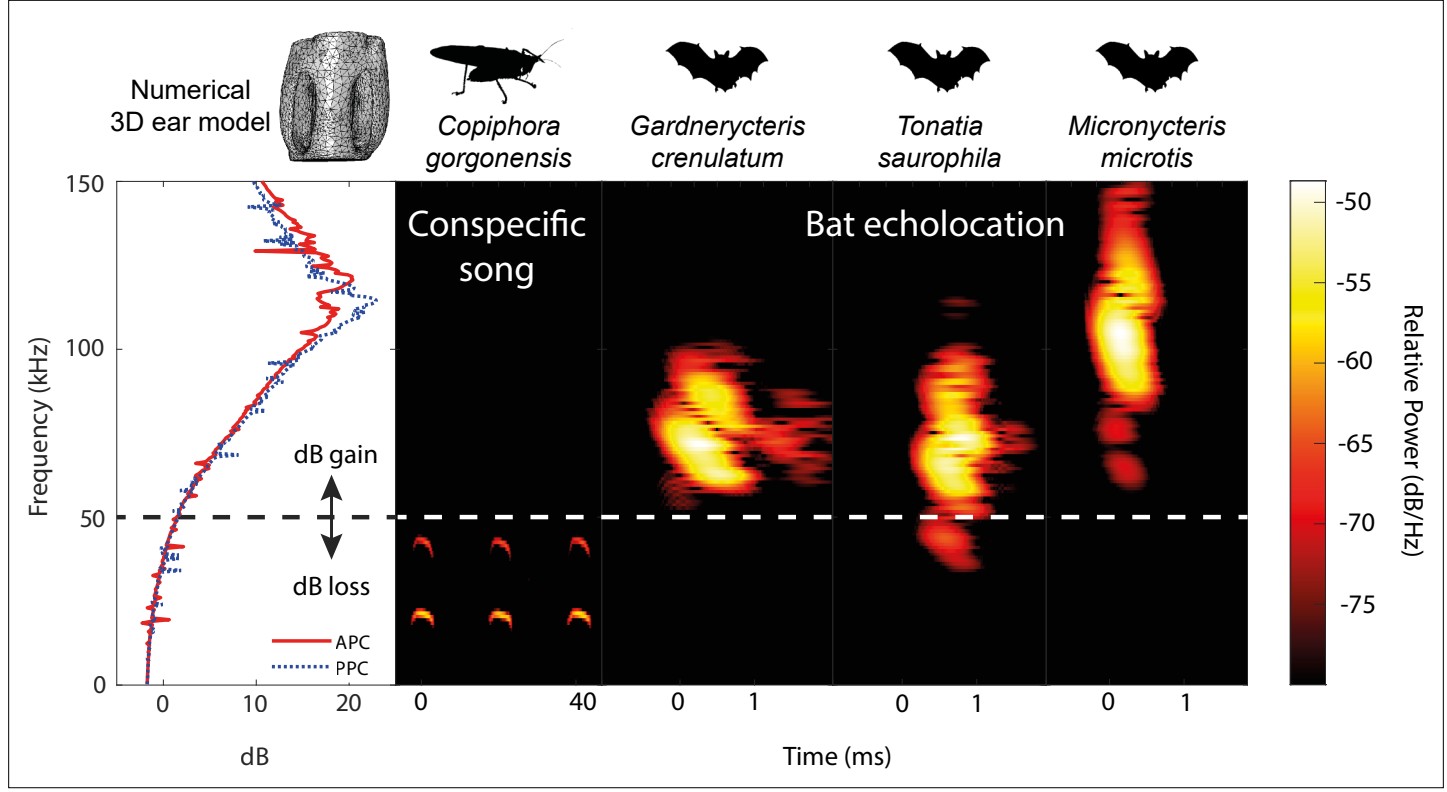

**Figure 6.** Ecological relevance of pinnae in *Copiphora gorgonensis*. Numerical results of sound pressure level gains (left subpanel) induced by the pinnae are present only at frequencies above c.a. 50 kHz, covering the range of echolocation frequencies of three native insectivorous gleaning bat species. The conspecific call of *C. gorgonensis* (dominant frequency and harmonics) on the other hand (dB$_{peak}$ at 23 kHz), is not enhanced by the presence of the pinnae (dB loss). Dotted line indicates the frequency at which gain = 0 dB. Spectrogram parameters: FFT size 512, Hamming window, 50% overlap; frequency resolution: 512 Hz, temporal resolution: 0.078ms. APC = anterior pinnal cavity, PPC = posterior pinnal cavity.

The online version of this article includes the following figure supplement(s) for figure 6:

**Figure supplement 1.** Comparison of acoustic behaviour of bats and katydids.

pressure within the cavities. Thus, the mechanical responses are different (in magnitude and dynamics) than sound pressure. In addition, the tympana of *C. gorgonensis* naturally resonate at ca. 23 kHz, the dominant frequency of the male calling song (*Celiker et al., 2020b*; *Jonsson et al., 2016*), and this was also observed in our experimental results, irrespective of pinnal presence or absence. Tympanal resonances around conspecific calling songs have been reported by early work on other species, some of whom concluded that both ear canal and pinnae resonated at the specific calling frequency (*Stephen and Bailey, 1982*). Whilst acknowledging the fact that technology at that time made it challenging to answer these questions, our results do not support this conclusion. Our results indicate that the pinnae, ear canals, and tympanal membranes exhibit different resonances. To test the influence of pinnal geometry alone on these ultrasonic gains, we printed 3D-scaled ears to conduct acoustic experiments and scaled the sound wavelength accordingly. The mean resonance of the 3D printed models was found to be beyond the species calling frequency, and this was also supported by the numerical models (*Figure 4E*). In experiments with ablated pinnae, high frequency pressure gains were dramatically reduced in both experiments and simulations. A small resonance was observed in both the numerical simulations and 3D print models after pinnal ablation, caused by the defective full removal of the pinnal structures (*Figure 4D, F*).

At high ultrasonic frequencies (>60 kHz), the pinnae-enclosed tympanal membranes of *C. gorgonensis* show strong mechanical vibrations induced by the resonances of the pinnal cavities (*Figure 5A*). This suggests that pinnae enhance sound pressure gains at high frequencies. It was previously demonstrated that even minuscule tympanal displacements in *C. gorgonensis* create large displacements of the *crista acustica* (*Montealegre-Z and Robert, 2015*). Tympanal displacements are magnified in the *crista acustica* and auditory vesicle as the effect of the lever action imposed by the vibration of the

tympanum and tympanal plates (*Figure 1B*; *Montealegre-Z et al., 2012*). Insect mechanosensory auditory neurons are capable of detecting incredibly small mechanical displacements, down to 100 pm (*Windmill et al., 2007*), approaching the theoretical limits of sensitivity (*Bialek, 1987*). Therefore, the sound pressure gain induced by the pinnae at ultrasonic frequencies (>60 kHz; *Figure 3* and *Figure 5A*) should produce sufficient tympanal displacement to induce a response in the auditory receptors, without amplification by the ear canal. Electrophysiological recordings of the auditory nerve from our experiments show a significant neural response to a broad range of frequencies (23–120 kHz) and sound pressures (46–94 dB SPL; *Figure 5C* and *Figure 5—figure supplement 1*, B), demonstrating that *C. gorgonensis* can detect very high ultrasonic frequencies.

## Ear pinnae as ultrasound detectors

Many papers testing the auditory role of pinnae in katydids were inconclusive, and limitations of equipment meant that researchers focused on testing the tympanal organ's response to conspecific frequencies. *Autrum, 1940*; *Autrum, 1942*; *Autrum, 1963* based his theory of the role of the pinnae in directional hearing on the assumption that sound acts only on the outer surface of the two tympana and did not consider the effect of sound entering the acoustic spiracle and ear canal, which was shown later, by other authors, to be the main source for acoustic orientation. Here we argue in support of *Lewis, 1974b* original observations that the pinnae in katydid ears act as ultrasound detectors. However, it is likely that some katydids do not use spiracular inputs, and that conspecific localization and predator detection depend solely on the external input (see below).

Power transmittance of ultrasonic frequencies suffers significant attenuation due to the high reflectance of sound waves along narrowing tubes (*Rossing and Fletcher, 1995*). The ear canal of *C. gorgonensis* and many other katydids has finite horn properties, which causes a drop in the gain above 60 kHz as reflections interfere (*Hoffmann and Jatho, 1995*; *Celiker et al., 2020a*). Therefore, the high variation in ear canal morphology in the katydid family (*Bailey, 1993*; *Bailey, 1990*) means that it is not always the primary input to the tympanal organ. High-frequency Pseudophyllinae katydids exhibit very small spiracles, and various forms of cuticular pinnae (*Morris et al., 1989*; *Bailey, 1993*). These insects have been shown to depend more on the external input than the internal for communicating with conspecifics (*Mason et al., 1991*). There appears to be a tendency that the dominant input for hearing has the larger opening for sound, at either the pinnal slit (external) or spiracle (internal). For example, the relationship between the external and internal openings dictates the principal auditory input in the ultrasonic hearing rainforest pseudophyllines *Myopophyllum speciosum*, *Haenschiella ecuadorica* and *Typophyllum* nr *trapeziforme*. In *C. gorgonensis*, the acoustic spiracle is large, naturally open and on average three times larger than the total area of the pinnal slits (1 mm$^2$ : 0.3 mm$^2$) which is inversely related to the general scale of pseudophylline ears. We propose that in species with large acoustic spiracles and pinnae, the pinnae evolved to increase the hearing range of the ear at extreme ultrasonic frequencies.

Pinnal asymmetry produces different resonances in the pinnal cavities, and these are seen in the spectrum of tympanal vibrations (*Figure 5A*), as well as and in the time domain recordings (*Figure 4C*). We do not know if these differences were related to different mechanical properties of the tympanal membranes, or to the area on each tympanal membrane available for positioning the laser beam through the slits. However, by concentrating ultrasonic frequencies into the pinnal cavity, the pinnae enhance ultrasonic reception of incidental sounds. The cavity-induced pressure gains are the product of the geometry of the pinnal slit in relation to the geometry and volume of the cavity (*Table 2*). Although the tympanal resonances are not strong at ultrasonic frequencies, these imparted forces are magnified by the lever action of the tympanum. The resonances afforded by the pinnal structures are evident as both the numerical and 3D print models do not include a vibrating tympanum. In *C. gorgonensis*, irrespective of incident sound pressure magnitude, the cavities provide a consistent pressure gain of at least 23 dB within the frequency range 100–120 kHz (*Figure 4C*). This is in contrast to tympanate moths that depend on the incident sound intensity for mechanical tuning of high frequency bat calls (*Fullard, 1984a*; *Fullard, 1984b*; *Windmill et al., 2006*) to produce gains up to 16 dB (*Fullard, 1984a*).

In *C. gorgonensis*, the dual inputs of the spiracle and the four external inputs function as a frequency range compensation system. As previously shown for *C. gorgonensis* (*Celiker et al., 2020a*), and in other species with large acoustic spiracles (*Heinrich et al., 1993*; *Michelsen et al., 1994*), the ear

canal with its finite horn geometry acts as a highpass filter, but with limited capabilities in providing pressure gains to high ultrasonic frequencies (<60 kHz, for *C. gorgonensis*) (*Lewis, 1974a*). In *C. gorgonensis* the ear canal enhances detection of the conspecific carrier frequency. While the reduction in sound velocity within the ear canal (~16% delay in *C. gorgonensis*, *Veitch et al., 2021*) contributes exceptional binaural directional cues, the external input provides in-real-time sensitivity to exploit fading bat ultrasounds, and to detect incoming bats from the onset of the echolocation sweep. Hence the ear canal is a less efficient method of bat detection as the angle of incidence and the reduction of sound velocity could delay reaction times and obfuscate the localisation of the predator. This suggests that katydids without pinnae should either exhibit other strategies for ultrasound detection, such as in the ear canal morphology, or not require the detection of ultrasounds exceeding ~60 kHz.

## Bat detection by katydid ear pinnae

Katydids form a key part of the diet of many insectivorous bat species worldwide (*Arlettaz et al., 1993*; *Buchler and Childs, 1981*; *Davison and Zubaid, 1992*; *Fenton and Royal Ontario Museum, 1975*; *LaVal and LaVal, 1980*; *Raghuram et al., 2015*; *Whitaker and Black, 1976*; *Zhang et al., 2005*). However, such ecological interactions have been more intensively studied in the Neotropical regions. Gorgona Island, Colombia, is home to over 33 bat species including at least three substrate gleaning bats of the neotropical leaf-nosed bat family Phyllostomidae (*Murillo et al., 2014*). The habitat of *C. gorgonensis* is in cluttered vegetation of the tropical forest understory (*Montealegre-Z et al., 2014*). In such environments, acoustic signals are heavily attenuated (*Romer and Lewald, 1992*), which leads to significant transmission loss (*Rheinlaender and Romer, 1986*; *Wiley and Richards, 1978*). Nevertheless, insects have evolved a variety of sophisticated receivers to perform call discrimination in these acoustically challenging environments (*Römer, 1993*). Neotropical katydids evolved behavioural and hearing strategies for survival against substrate gleaning bats (*Belwood, 1990*; *Belwood and Morris, 1987*; *Nickle and Castner, 1995*; *ter Hofstede et al., 2010*; *Ter Hofstede et al., 2017*). Acoustic adaptations by katydids to evade bat predation include the use of narrow bandwidths (tonal calls), high carrier frequencies, and sporadic calling in order to diminish signal proliferation in the environment (*Belwood and Morris, 1987*; *Morris et al., 1994*; *Morris and Beier, 1982*; *Rentz, 1975*; *Heller, 1995*). Certain adaptations are a trade-off as the katydid becomes more conspicuous and vulnerable to other predators as the communication method changes. For example, katydids that perform vibrotaxis can likely attract spiders, scorpions (*Robinson and Hall, 2002*) and primates, as well as bats (*Geipel et al., 2020*). Likewise, bats foraging in the cluttered understory also face similar acoustic shortcomings, affecting their echolocation abilities (*Page et al., 2020*; *Geipel et al., 2020*). Thus, several phyllostomid substrate gleaning bats are very well adapted to hear prey-produced cues like rustling noises or mating calls, including those of male katydids (*Belwood and Morris, 1987*; *Falk et al., 2015*; *Geipel et al., 2021*). At least one common gleaning bat species, *Micronycteris microtis* (Phyllostomidae), uses a sophisticated echolocation strategy to detect katydids concealed in vegetation (*Geipel et al., 2019*; *Geipel et al., 2013*). Despite their passive acoustic defences, calling from sheltered locations and being equipped with very large mandibles and sharp fastigia, katydids like *C. gorgonensis* are predated by phyllostomid bats (*Ter Hofstede et al., 2017*).

Our numerical and experimental evidence suggests that the greatest ultrasonic gain of the pinnae is at resonances matching the frequency range of the echolocation calls of native gleaning bats (*Figure 6*). As neotropical gleaning bats approach their target, they emit short, broadband, multi-harmonic sweeps, demodulate the frequency from higher frequencies above 135 kHz to as low 35 kHz (*Geipel et al., 2021*; *Yoh et al., 2020*). In terms of predator detection, a katydid like *C. gorgonensis* has an excellent chance of detecting the calls of a hunting bat at the start of the sweep. Responses to these high frequencies are supported by LDV recordings of tympanal motion in intact ears, and audiograms that show a broad mechanical, behavioural, and neural response to ultrasonic frequencies (*Figure 5A–C*; *Table 3*). A gain of 16–20 dB at the start of the bat call provides essential awareness time [(≤0.86ms in terms of duration of the complete sweep (*Geipel et al., 2021*)] to *C. gorgonensis* as a result of the tympanal pinnae. This demonstrated acute sensitivity (or predator escape response) to frequencies matching both the pinnal cavities and the call of echolocating bats. The low/flat behavioural threshold at high frequencies between 90 and 120 kHz, has been reported for other species. The average startle behavioural threshold in *C. gorgonensis* was 59.28 ± 1.80 dB SPL

(*Figure 5B*), which is comparable to the behavioural response in *Neoconocephalus ensiger* (*Faure and Hoy, 2000*).

These broad responses to ultrasound are common in several pinnae-bearing katydid subfamilies of Tettigoniidae. Early and more recent researchers, obtaining recordings from the tympanal nerve and the T-cell in several katydids bearing auditory pinnae [e.g., Pseudophyllinae and Conocephalinae species] (*Wever and Vernon, 1959*; *Faure and Hoy, 2000*; *Deily and Schul, 2006*; *Schul and Patterson, 2003*; *ter Hofstede et al., 2010*), Tettigoniinae (*Autrum, 1940*; *Rheinlaender and Romer, 1986*), showed a broad sensitivity in the range 5–100 kHz. In addition, katydid species living in sympatry with *C. gorgonensis* like *Supersonus aequoreus* (the most ultrasonic katydid found in nature to date *Sarria-S et al., 2014*), *Ischnomela gracilis*, and *Eubliastes aethiops* exhibit similar cavity-induced pressure gains in the range of phyllostomid echolocation calls (*Figure 6—figure supplement 1A*).

The pressure – time difference receiver of many katydids is a unique system that can capture different ranges of frequencies between the multiple entry inputs that can obviate the limitations of each but is also capable of compensating for limitations in auditory orientation (*Michelsen et al., 1994*; *Veitch et al., 2021*). For katydids, incident sounds from elevation are difficult to perceive (*Römer, 2020*). Hence, the ability of the ears to be physically positioned and rotated in accordance with the movement of the foretibial leg joints (*Autrum, 1940*; *Autrum, 1963*) permits the ear to hear elevated sounds. For ultrasonic reception, a total of four external inputs (left and right anterior and posterior tympana) plus the sub-slit cavities asymmetrically recessed to the distal end, may be behaviourally articulated to enhance the detection of bats calling from elevated positions toward the katydids. The physical separation between the external inputs of each ear should yield sufficient binaural cues, and merits further investigation.

## Ideas and Speculation: Katydid ear pinnae and the fossil record

The presence of ear pinnae in katydids in the fossil record is known from late Eocene (*Gorochov, 2010*), but has been neglected. Katydid ancestors (e.g. Haglidae and Prophalangopsidae from Upper Jurassic; *Gu et al., 2012*; *Plotnick and Smith, 2012*) and early katydids (Tettigoniidae) from the middle Paleogene (early Eocene; *Greenwalt and Rust, 2014*; *Rust et al., 1999*) all show naked tympana without pinnae (likely the plesiomorphic condition). Auditory pinnae may have evolved as a relatively recent apomorphic character in the family Tettigoniidae for more sophisticated hearing in bat detection. The earliest echolocating bats are from the early Eocene, ~55 mya (*Teeling et al., 2005*). The fossil record places a potential emergence of pinnae some 40–44 mya (*Gorochov, 2010*). Analogous ear pinnal adaptations are observed in some Eneopterinae crickets (tribe Lebinthini) (*Schneider et al., 2017*), which differ from field crickets in their use of high frequencies for specific communication (12–28 kHz). These crickets also emerged in the Eocene (*Vicente et al., 2017*) and while their ancestors exhibit only one (posterior) functional tympanum, the extant forms show two functional, asymmetric tympana, with the anterior tympanum covered by pinnae (*Schneider et al., 2017*). Such adaptations suggest a new paradigm of the dual role of the ears, in detecting conspecific and bat echolocation calls. As a working hypothesis, we propose that ear pinnae have a unique origin across the ca. 8,100 living species of Tettigoniidae (*Cigliano et al., 2021*) in response to the emergence of bats during the early Eocene, and that it was subsequently lost or modified several times.

Although katydid ear pinnae have never been mapped in the most recent molecular phylogenies (*Song et al., 2020*; *Mugleston et al., 2018*; *Mugleston et al., 2013*), we observe a potentially unique origin of ear pinnae in the family Tettigoniidae, with multiple losses or retrogressions in modern species, including the large subfamily Phaneropterinae, and the Mecopodinae, predominantly known to have naked tympana. Comparative analyses using large phylogenies are in progress to solve this working hypothesis. While little is known about the species-specific ecologies and life histories of the Phaneropterinae and Mecopodinae, it would not be surprising that, without pinnal structures, some species evolved sophisticated ear canals with exceptional broadband response for bat detection (*Heinrich et al., 1993*; *Hoffmann and Jatho, 1995*; *Michelsen et al., 1994*; *ter Hofstede et al., 2010*). This implies that some non-pinnae-bearing species could have a unique ear function via the ear canal, which can detect conspecific calls as well as bats. It could also be that many species have evolved diurnal activity patterns in response to bats (*Fornoff et al., 2012*; *Heller and von Helversen, 1993*).

Other adaptions involve dwelling in dense vegetation that challenges hunting bats (*Lang and Römer, 2008*). Katydids like *Conocephalus* spp. and *Orchelimum* spp. with tympanal pinnae are mostly active during the daytime, and a majority dwell in dense meadows. Their calling songs are of unusual broadband energy, in many species expanding above 60 kHz (*Wever and Vernon, 1959*; *Fullard et al., 1989*). In this case, the retention of pinnae might assist in conspecific directional hearing, permitting enhanced acoustic ranging (*Harness and Campbell, 2021*) in such dense grass environments. The functional and ecological significance of pinnae across the Tettigoniidae is likely to provide a rich avenue for future biophysical research.

## Materials and methods

### Specimens

*Copiphora gorgonensis* (Tettigoniidae: Copiphorini) is endemic to Gorgona National Natural Park, Colombia (02°58′03″N 78°10′49″W). The original generation of the species were imported to the UK under the research permit granted by the Colombian Authority (DTS0-G-090 14/08/2014) in 2015. The specimens were ninth generationfrom captive bred colonies maintained at 25 °C, 70% RH, light: day 11 h: 23 h. They were fed ad libitum diet of bee pollen (Sevenhills, Wakefield, UK), fresh apple, dog food (Pedigree Schmackos, UK) and had access to water. Live experiments were conducted on seven adults of *C. gorgonensis* from our laboratory breeding colonies at the University of Lincoln (Lincoln, UK). Following experimentation, these specimens plus an additional four females already stored in ethanol were micro-computed tomography scanned for finite element modelling; totalling 17 ears (10 female, 7 male). Live specimens were subsequently preserved in 100% ethanol-filled jars and stored in a freezer at –22 °C at the University of Lincoln.

### Simultaneous recordings of tympanal vibrations using laser Doppler Vibrometry

Insects were chemically anesthetized using triethylamine-based agent FlyNap (Carolina Biological Supply, USA) for 15 min prior to the mounting process, and remained awake throughout the duration of the experiment. The animals were dorsally mounted using a specialized platform to isolate the external and internal sound inputs and also mimic their natural stance (*Figure 2—figure supplement 1*). A rosin-beeswax mix was used to fix the pronotum, and the mid- and hindlegs, to the mount. This specialized platform (*Jonsson et al., 2016*) consists of two Perspex panels (1.61 mm thick) that are joined by latex and suspended in the air by a 12 × 12 mm metal frame attached to a micromanipulator (World Precision Instruments, Inc, USA; see *Montealegre-Z et al., 2012*). At the Perspex junction, the forelegs of the insect were extended through arm holes cut in the Perspex and attached on a rubber block with metal clasps. A metal clasp was placed on each foretibia and forefemur (total of 4) to arrest foreleg motion. The arm holes and frame borders were sealed with latex to block sound propagation to the spiracle.

The laser Doppler vibrometry system consisted of a the OFV-2520 Dual Channel Vibrometer - range velocity controller for operating two single point laser sensor heads, (OFV-534, Polytec, Germany) each with VIB-A–534 CAP camera video feed and laser filters. Each sensor head was mounted on a two-axis pivoting stage (XYZ, Thorlabs Inc, USA) anchored to an articulating platform (AP180, Thorlabs Inc, USA) and manually focused at 10.5 cm above a vibration isolation table (Pneumatic Vibration Isolation Table with a B120150B - Nexus Breadboard, 1200 mm × 1500 mm × 110 mm, M6 × 1.0 Mounting Holes, Thorlabs Inc, USA) supported by an anti-vibration frame (PFA52507 - 800 mm Active Isolation Frame 900 mm × 1200 mm, Thorlabs Inc, USA) in an anechoically isolated chamber (AC Acoustics, Series 120a, internal dimensions of 2.8 m × 2.7 m × 2.7 m). The sensor heads were outfitted with magnification microscopic lenses (Mitutoyo M Plan 10× objective for Polytec PSV-500 single laser head OFV 534, Japan) and positioned about 35–40 mm away from the insect foreleg at 45° angles towards the Perspex surface (*Figure 2—figure supplement 2*). The narrow entrance to the pinnal cavities restricted the use of LDV, such that we could not measure tympanal vibrations across the entire membrane. Therefore, the placement of the sensor heads was limited to positions where they were perpendicular to the tympanum of interest. The sensor speeds were maintained at 0.005 m s$^{-1}$V$^{-1}$ and recorded using an OFV-2520 internal data acquisition board (PCI-4451; National Instruments, USA).

Tympanal vibrations were induced by a four-cycle sinusoidal wave at 23, 40, and 60 kHz. The closed-field configuration for a probe of the loudspeaker restricted the delivery of high ultrasonic stimuli to 60 kHz. A rotating automated stage (PRM1Z8 rotation mount, Thorlabs Inc, USA) with a KDC101 K-Cube DC Servo Motor Controller (Thorlabs Inc, USA) positioned a multi-field magnetic loudspeaker (MF1, Tucker Davis, USA) with a parabolic nozzle (see Supplementary Materials from *Veitch et al., 2021*) and plastic probe tip (3.5 cm L × internal diameter 1.8 mm W) about 3.5 mm away from the mounted insect and 10.2 cm above the breadboard table. The speaker was moved across a 12 cm semi-circle radius in 1° steps (0.56 mm). The probe tip was positioned at point zero and 20 single shot recordings at 1° intervals, totalling 10° at either side (*Figure 2—figure supplement 2*). A high quality 500 band pass filter was applied at 10–30 kHz for the 23 kHz recordings, 30–50 kHz for the 40 kHz recordings, and 50–70 kHz for the 60 kHz recordings. All acoustic signals were generated by a waveform generator (SDG 1020, Siglent, China), synchronized with the LDV, amplified (ZB1PS, Tucker Davis, USA) and measured by a 1/8″ (3.2 mm) omnidirectional microphone (B&K Type 4138, Brüel & Kjaer, Nærum Denmark) located about 3 mm from the tympanum. The microphone, with built in preamplifier (B&K Type 2670, Brüel & Kjær, Nærum, Denmark), was calibrated using a sound-level calibrator (B&K Type 4237, Brüel & Kjaer, Nærum, Denmark) and set to 316 mV/Pa output via a conditioning amplifier (Nexus 2690-OS1, Brüel & Kjær, Nærum, Denmark). A reference measurement was performed by placing the microphone 3 mm from the probe tip to the loudspeaker before each experiment. Using a micromanipulator, the microphone was positioned approximately 3–3.5 mm from the ear to monitor the acoustic isolation of the platform.

The sensor heads were manually focused on the external tympanal surface using the 2-axis pivoting stage and manual wheel with the aid of the sensor head camera output displayed on an LED screen. For the time measurements, the point zero was found for each leg and for each test frequency. The point zero was the point where the displacements from the anterior tympanal membrane and posterior tympanal membrane matched the oscillation phase of the generated four-cycle sinusoidal waves. This ensured that the vibrations of the tympanal membranes were synchronous relative to the speaker position. Displacement amplitudes from the same cycle order number were measured from each sensor head reference, and approximately 252 data points were measured per ear.

After recording the vibrations for both ears of the tested individual, the cuticular pinnae were carefully excised using a razor blade (taking care not to damage the tympanal organs or the fine layer of tissue ventrally connected to the tympanal membranes). The measurements were repeated for each ear following the same protocol.

Time and displacement measurements were analysed by identifying the second oscillation of the four-cycle tone generated waves in each software window (PSV 9.4 Presentation software, Polytec, Germany). Phase calculations were obtained using the equation $\varphi° = 360° \times f \times \Delta t$ where $f$ is frequency (kHz) and $\Delta t$ (ms) the difference in arrival times between the anterior and posterior tympana.

## Anatomical measurements of the external tympanal input

To produce 3D data for modelling, 17 ears of *C. gorgonensis* were scanned using a SkyScan 1172 X-ray micro-computed tomography scanner (Bruker Corporation, Billerica, MA, USA) with a resolution between 1.3 and 2.9 µm (55 kV source voltage, 180 µA source current, 300ms exposure and 0.1° rotation steps). As experimental procedures required removal of the cuticular pinnae, eight additional specimens with intact pinnae were scanned. The micro-computed tomography projection images were reconstructed with NRecon (v.1.6.9.18, Bruker Corporation, Billerica, MA, USA) to produce a series of orthogonal slices. The 3D segmentation of the ear, measurements of the ear cross section and width, and volumetric measurements of the pinnal cavities were performed with the software Amira-Aviso 6.7 (Thermo Fisher Scientific, Waltham, Massachusetts, USA). Micro-computed tomography stereolithography files (STL) were generated for numerical modelling using established protocols (*Jonsson et al., 2016*; *Veitch et al., 2021*) and to 3D print ear models.

For 2D measurements of the cavity slit area, pinnal protrusion, and the distance between the pinnal cavities, an Alicona InfiniteFocus microscope (G5, Bruker Alicona Imaging, Graz, Austria) at 5× objective magnification was used to capture images of collection specimens with intact pinnae, with a resolution of about 100 nm (*n* = 8 ears).

## 3D printed model time and frequency domain measurements of pinnal cavities

For time domain measurements, 3D models of the ears (n = 8; 1 male and 1 female ± pinnae) were placed on a micromanipulator arm with blu-tac (Bostik Ltd, Stafford, UK) and positioned frontally 30 cm from a MF1 loudspeaker at the same elevation. A 25 mm tipped B&K Type 4182 probe microphone (Brüel & Kjær, Nærum, Denmark) with a 1 × 25 mm (0.99") probe tube length and 1.24 mm (0.05") interior diameter, calibrated using a B&K Type 4237 sound pressure calibrator was placed ventral to the ear. The ear moved on the microphone using an electronic micromanipulator (TR10/MP-245, Sutter Instrument, Novato, California, USA), to a position 1 cm from the back of the cavity. Stimuli delivered were individually scaled to match the wavelength of a real-size ear (e.g. for a 1:10 scale printed model, the frequency delivered to simulate 120 kHz would be 120/10 = 12 kHz) to account for variation in printed model scaling. 3D printed models were scaled 1:11.43 (male 1:11.33; female 1:11.53) with the corresponding average scaled stimuli of 2.01 kHz for 23 kHz, 3.50 kHz for 40 kHz, 5.25 kHz for 60 kHz, and 9.63 kHz for 110 kHz. Four cycle pure tones were produced using the function generator, and the amplitude set to deliver 1 Pa to the microphone at each frequency. Received signals were amplified using a B&K 1708 conditioning amplifier (Brüel & Kjær, Nærum, Denmark), and acquired using a PSV-500 internal data acquisition board at a sampling frequency of 512 kHz. The microphone remained stationary during the experiments, nor was its direct path to the speaker obstructed. Instead, the microphone entered the ear via a drilled hole, allowing the pinnae to surround the tip of the microphone. Thus, the reported sound pressure gains result solely from the cavities of the 3D model, and not the motion of the microphone. When the microphone was positioned inside the cavities, the gap between the drilled hole and microphone probe was sealed with blu-tac to mimic the real cavity and avoid acoustic leaking (see *Videos 1 and 2*).

To calculate the frequency that produced the best gain, the MF1 loudspeaker was replaced with a RAAL 140-15D Flatfoil loudspeaker (RAAL, Serbia), with a different amplifier (A-400, Pioneer, Kawasaki, Japan). This speaker was able to deliver a broadband stimulus of periodic chirps, generated within Polytec 9.4 software, with a simulated frequency range of 2–150 kHz. Recording in the frequency domain, at a sampling frequency of 512 kHz, the amplitude of the broadband stimulus was mathematically corrected within the software to deliver 60 dB at all frequencies. The reference frequency spectrum with no ear present could be subtracted from the frequency spectrum reported within the cavities to calculate frequency-specific gain and thus cavity resonance. Gain was calculated by subtracting the probe microphone sound pressure (dB) measured 1 cm outside of the cavity from inside the pinnal cavity measurements (*Figure 3*; *Video 1*).

For comparative purposes, the ears of the following sympatric and pinnae-bearing katydid species from Gorgona Island were also 3D printed and subjected to experiments according to the aforementioned protocol: *Ischnomela gracilis*, *Supersonus aequoreus* and *Eubliastes aethiops* (see *Figure 3—figure supplement 1*). Frequency domain recordings of the cavity resonance, and time domain recordings of pure tone gains were then exported as .txt files for analysis.

To 3D print the ears of *Copiphora gorgonensis*, *Ischnomela gracilis*, *Supersonus aquoreus* and *Eubliastes aethiops*, micro-CT stereolithography files (STL) were imported into the software CHITUBOX 64 (Chitubox, Guangdong, China). The models were scaled to be approximately 12× larger than the actual ears. Support structures and a base printing platform were then added to support the model, with a 0.2 mm attachment thickness to the model. Supported models were delivered via USB to a Mars Elegoo Pro 2 3D Printer (Elegoo Inc, Shenzhen, China). Models were printed using grey ABS-like photopolymer resin (exposure parameters: 20 s first layer, 5 s normal layers) with a solidification wavelength of 405 nm. When printing was complete (about 1 hr 30 min), models were washed in 100% isopropyl alcohol, rinsed in cold water, then exposed to UV light in an Elegoo Mercury Plus curing station (Elegoo Inc, Shenzhen, China) for 8 min. To prepare the models for entry of the probe microphone into the pinnal cavities, 2 mm diameter holes were drilled into the centre of the base of each cavity (*Figure 3*).

## Numerical modelling

The mathematical models have been constructed as a scattering acoustic – structure interaction problem and simulate the acoustic response of the pinnal cavities to an incident plane acoustic wave in an air domain. Hence, the 3D model considers the interaction of the sound wave with the ear, for

which realistic material properties have been incorporated. The air acoustic domain is truncated as a sphere with a 3 mm radius that is centered around the ear (*Figure 4—figure supplement 1A*). Two different geometries of the ears were taken as part of the mathematical model domain: pinnae intact and pinnae removed (*Figure 4—figure supplement 1B*).

The models were considered both in the frequency and the time domains, and were solved using the acoustic-shell interaction module of the software Comsol Multiphysics, v5.6 (*Comsol, 2021*). For the frequency domain models, the incident wave was taken to be a chirp with an amplitude of 1 Pa and frequency 2–150 kHz, directed at point zero as defined in the in the section *vibrational measurements*. For the time domain models, three different incident waves were used, with amplitudes 1 Pa and frequencies 23, 40, 60 kHz. The direction of the waves was taken as –10°, –5°, 0°, 5°, and 10° on a fixed plane perpendicular to the ear, with 0° corresponding to point zero.

For the numerical simulation of the problem, we solved a system of equations representing the sound pressure (SPL dB) inside and around the *C. gorgonensis* ear, resulting from the interaction of the ear with an incident plane acoustic wave in an air domain. The air acoustic domain is truncated as a sphere with a 3 mm radius that is centered around the ear (*Figure 4—figure supplement 1A*).

Two different sets of mathematical models were considered in the described geometry, within the frequency and the time domains. For the frequency domain calculations, the solution to the Helmholtz equation

$$\frac{1}{\rho}\Delta p_f + k^2 p_f = 0 \tag{1}$$

was considered for the acoustic system, where the parameters $\rho$ = is the density of air, $k = \omega/c$ is the wavenumber, $\omega$ is the angular frequency and $c$ = 343 m s$^{-1}$ is the speed of sound in air. The variable $p_f(x)$ is the total pressure in the frequency domain, which is dependent on the 3D spatial variables $x = (x, y, z)$, and $\Delta = \frac{\partial^2}{\partial x^2} + \frac{\partial^2}{\partial y^2} + \frac{\partial^2}{\partial z^2}$ is the Laplace operator.

At the outer perimeter of the sphere, to allow for a radiated or scattered spherical wave to travel out of the modelling domain without reflections, a spherical radiation boundary condition was applied in the following form:

$$n.\nabla p_f + \left(ik + \frac{1}{r}\right)p_f - \frac{r\Delta_{\|p_f}}{2(ikr+1)} = n.\nabla p_{fi} + \left(ik + \frac{1}{r}\right)p_{fi} - \frac{r\Delta_{\|p_{fi}}}{2(ikr+1)} \tag{2}$$

where $n$ is the normal vector, $r$ is the distance from the source location, the operator $\Delta_{\|}$ denotes the Laplace operator in the tangent plane at a particular point and $i = \sqrt{-1}$. This boundary condition was based on an expansion in spherical coordinates given in *Bayliss et al., 1982* and implemented to the second order. The right-hand side of *equation (2)* allows for an incoming plane wave defined as

$$p_{fi} = e^{-ik\left(\frac{x.e_k}{\|e_k\|}\right)}$$

with magnitude 1 Pa and frequency ranging from 2 to 150 kHz. The wave travels from the direction $e_k$, which was taken as normal to the front of the ear (point zero).

The ear itself was considered as an isotropic shell system which allowed for the calculation of displacement and stresses resulting from the fluid load. The tympanal membranes were defined as a shell made of a homogeneous, linear elastic material with a Young's modulus of 2 GPa, density of 1300 kg/$m^3$, Poisson's ratio of 0.3, and thickness 5 μm (*Montealegre-Z and Robert, 2015*; *Figure 4—figure supplement 1B*). The rest of the ear was assumed to have a thickness of 175 μm and the same material properties as the tympana.

Finally, the continuity between the acoustic and shell systems was retained by accounting for the interaction between the two systems. After calculating the frequency response of the ear to the fluid load in the form of harmonic displacements and stresses, the model used the displacement magnitude of the solid surface in the acoustic domain inner boundary to ensure continuity. This is represented by the equations

$$n.\frac{1}{\rho}\nabla p_f = \omega^2 U_{sf},$$

$$F_{Af} = p_f n,$$

At the intersection of the ear with the sphere, where $U_{sf}$ is the ear (shell) displacement and $F_{Af}$ is the load (force per unit area) experienced by the shell structure.

An analogous model was also considered in the time domain, for which instead of **equation (1)**, the wave equation

$$c^2 \Delta p_t = \frac{\partial^2 p_t}{\partial t^2}$$

was solved for in the acoustic domain, where $p_t(x, t)$ is the total pressure in the time domain, which is dependent on both the space variables $x$ and the time variable $t$. The boundary condition (2) was also replaced by the time dependent spherical wave condition

$$n.\nabla p_t + \left( \frac{1}{c} \frac{\partial p_t}{\partial t} + \frac{1}{r} p_t \right) = n.\nabla p_{ti} + \left( \frac{1}{c} \frac{\partial p_{ti}}{\partial t} + \frac{1}{r} p_{ti} \right),$$

where the incident wave $p_{ti} = sin\left( 2\pi f_0 \left( t - \frac{x.e_k}{c\|e_k\|} \right) \right)$, at frequencies $f_0 = 23, 40$ and $60\,kHz$. Finally, the continuity of the acoustic and shell systems was ensured with the equations

$$n.\frac{1}{\rho} \nabla p_t = \frac{\partial U_{st}^2}{\partial t^2},$$

$$F_{At} = p_t n,$$

at the intersection of the ear with the sphere, where $U_{st}$ is the time dependent displacement of the ear and $F_{At}$ is the time dependent load experienced by the shell structure.

The numerical solution to the problem was obtained using the finite element method for the spatial variables in both the time and frequency domain simulations. For forming the finite-element mesh, the maximum diameter used for the tetrahedral elements in the sphere was $h_{max} = \frac{c}{6 \times f_0}$, where $c = 343 m/s$ and $f_0 = 150 kHz$ (**Figure 4—figure supplement 1B**, B). Hence, even at the largest frequency considered, there were six tetrahedral elements per wavelength. Quadratic Lagrange elements were applied for the solution.

For the time domain solution, the time variable was solved for using the Generalized alpha method, with a constant time step of $\Delta t = \frac{1}{60 \times 150} s$, so that the Courant-Friedrichs-Lewy (CFL) condition (**Courant et al., 1967**), defined as $CFL = \frac{c \times h_{max}}{\Delta t}$ was 0.1, which gives a reliable approximation of the solution.

## Tympanal response to broadband stimulation

For the tympanal tuning measurements, we exposed seven specimens (4 males, 3 females) to free field broadband (periodic chirp 20–120 kHz) stimulation presented by an ipsilaterally positioned SS-TW100ED Super-Tweeter (Sony, Tokyo, Japan) with a 20 kHz built-in high-pass filter using an Avisoft Bioacoustics Ultrasonics Power Amplifier (Avisoft Bioacoustics, Glienicke/Nordbahn, Germany). A rosin-beeswax mix was used to fix the pronotum, and the mid and hind legs, to the mount (see **Montealegre-Z et al., 2012**) after the insects were chemically anesthetized using FlyNap. Insects were then elevated to the same level as the LDV and positioned 15 cm from the loudspeaker. A 1/8″ B&K Type 4138 microphone was placed about 3 mm in front of the ear of interest and recorded the stimulus. Mechanical responses were acquired using a PSV-500 internal data acquisition board at a sampling frequency of 512 kHz. The amplitude was corrected to maintain 60 dB SPL at all frequencies. Data was collected as magnitude (velocity/sound pressure).

## Behavioural audiograms

Behavioural audiograms were measured from nine tethered female (*n* = 9) *C. gorgonensis* to test behavioural response thresholds to controlled auditory stimuli (20–120 kHz). Specimens were tethered from the pronotum to control for a constant position sound pressure, while the specimen walked on a foam rotating cylinder. The cylinder (15 cm diameter × 15 cm deep) was customised by the Foam Superstore. The cylinder freely rotated on a rod crossing along its longitudinal axis, with each end resting on the centre of a Hard Disk Drive Spindle Wheel (custom designed using parts of old computer hard drives). These wheels produce smooth rotation of the rod and cylinder that do not disturb the insect. Specimens were glued from the pronotum to a 25 cm wooden rod (4 mm diameter) using bees wax (Fisher Scientific UK, Limited, Leicestershire, UK) and colophony resin (Sigma-Aldrich

Co. St. Louis, MO, USA; Product No. 60895–250 G) in a 1:1 mix. The wooden rod was held by a micromanipulator which allows positioning of the insect on the rotating foam cylinder. Each specimen was left to adapt to the new situation for 15 min, before the experiment started. This experimental setup was mounted on a Pneumatic Vibration Isolation Table (B120150B) supported by an anti-vibration frame (PFA52507). All experiments were conducted inside an acoustic booth (AC Acoustics, Series 120a, internal dimensions of 2.8 m × 2.7 m × 2.7 m). A disadvantage of the treadmill used here was that the insect is forced to walk in the forward direction, different to other more sophisticated air-cushioned spherical treadmill systems that allow movement in any direction (*Hedwig and Poulet, 2004*; *Mason et al., 2001*). However, since we were not interested in directional responses, but only on startle behaviour, this simple treadmill was useful.

Acoustic stimuli were generated in a function generator (Agilent 33120 A, 15MHz Function/Arbitrary waveform generator, Agilent Technologies UK Ltd., Edinburgh, UK), and shaped into 10 ms pulses (2 ms linear rise/fall) at 6 volts peak-to-peak. Function generator output was connected into a portable single channel ultrasonic power amplifier suited for the ultrasonic speakers, Model B without 200 V bias voltage generator (Avisoft Bioacoustics, Glienicke/Nordbahn, Germany). Sound stimulus was delivered using a SS-TW100ED Super-Tweeter loudspeaker, which has a frequency response in the range 20–125 kHz. The input from the Avisoft amplifier was high-pass filtered at 20 kHz using the built-in filter of the Sony Tweeter. The speaker was positioned 15 cm antero-lateral from the specimen. The amplitude of the stimulus was monitored using a B&K 1/8" precision pressure Type 4138 microphone a preamplifier (B&K model 2633, Brüel & Kjær, Nærum, Denmark). The microphone was calibrated using a sound level calibrator (B&K Type 4231), and positioned 5 cm above of the tethered insect. The acoustic stimuli were constantly monitored in real time using the analyser window of the Polytec laser software.

At each frequency (20 : 5 : 120 kHz), pure tones of 10ms duration were played at increasing amplitude (40 : 5 : 90 dB SPL) to measure behavioural thresholds of the nine female *C. gorgonensis*. Starting at the lowest amplitude for a given frequency, each stimulus lasted 1 s, and consisted of ten 10 ms pulses presented at a rate of 100 Hz. Three types of behaviours were observed: (1) interruption of walking; (2) alert (the katydid tried to jump or adopted a defensive position); (3) no response. If any of reactions (1 and 2) occurred, the stimulus was decreased by 5 dB and the animal was re-tested once walking resumed. Threshold was defined as the lowest amplitude that reliably elicited a behaviour and for each given frequency. We anticipated that above this sound pressure, the insect continued hearing the stimulus. If no response occurred, the stimulus was repeated (after a few seconds of silence) to verify the lack of response. If still no response, the stimulus amplitude was increased by 10 dB steps and the katydid was re-tested.

For purposes of analysis, for each specimen the threshold at each frequency was annotated in a matrix for further calculation of mean vector and standard deviations. Not all specimens showed consistent response at all frequencies and treatments, and if no response was shown to a particular stimulus, but the specimen was shown response to other stimuli, the missing response was entered as NaN (missing value identifier for Matlab matrix computation; see *Table 3*).

## Neural audiograms

Suction electrode recordings were obtained from the auditory nerves of five adult *C. gorgonensis* following previously described methods (*Isaacson and Hedwig, 2017*). Briefly, animals were restrained dorsal side up in plasticine with their acoustic spiracles and tympana exposed to the air. One auditory nerve was sampled per animal, which was accessed by removing a small window of cuticle from a front femur and dissecting away any obstructing material. A pre-prepared polycarbonate electrode (1 mm outer diameter; 0.5 mm internal diameter; pulled by hand over a soldering iron and cut to a terminal internal aperture of ~40 μm) was filled with HEPES-buffered saline that had been made viscous with 4% Tylose H200 NP2 (ShinEtsu, Wiesbaden, Germany) to prevent leakage from the tip. The electrode was fitted into a custom-made holder, with a platinum wire inserted into the saline. The electrode tip was then placed onto the auditory nerve using a micromanipulator, and sealed using gentle suction. A platinum reference electrode was inserted into a small incision in the distal tibia.

Whole-nerve activity in response to sound was recorded using a differential amplifier (model 1700, A-M Systems Inc, Carlsborg, WA, USA), and sampled at 15 kHz using an analogue-to-digital converter and recording software (CED Micro 1401 and Spike2 version 7, Cambridge Electronic Design,

Cambridge, UK). Acoustic stimuli were created by a function generator (SDG1020, Siglent Technologies, Augsburg, Germany), consisting of a fully amplitude modulated pulse with a cycle frequency of 1 Hz (0.5 s ON, 0.5 s OFF). This signal was carried via a power amplifier (SA1, Tucker-Davis Technologies System, Alachua, FL, USA) to an ultrasonic power amplifier (Avisoft Bioacoustics, Glienicke/ Nordbahn, Germany). From here, the signal passed through a Sony Tweeter capable of producing acoustic signals from 20 to 125 kHz positioned 15 cm from the animal, with a clear path to both the tympana and acoustic spiracle of the recorded ear. To calibrate the SPL of the signal, a B&K Type 4138 1/8" condenser microphone with built-in pre-amplifier (B&K Type 2670) was connected to the same data acquisition system as the neural recording via a power amplifier (Type 12AA, G.R.A.S., Holte, Denmark). From here, the signal amplitude was calibrated at 94 dB SPL (1 Pa) using a portable sound pressure calibrator (B&K Type 4231). The microphone was then placed above the tympanal organ, and the SPL of the stimulus modified until the output SPL was equal to the calibrated 94 dB SPL. To modify the SPL following calibration, the gain output of the SA1 power amplifier was reduced in –6 dB steps. Eleven different sound stimuli consisting of pure tones ranging from 23 to 120 kHz were randomly presented. Each stimulus was presented nine times per frequency at increasing sound pressures from 46 to 94 dB in 6 dB increments (giving a total of 10 repeats ×9 sound pressures ×11 frequencies = 990 responses per animal).

Recordings were digitized using a Micro1401 mk II (Cambridge Electronic Design (CED), Cambridge, UK) for observation and storage for later analysis on a PC using Spike2 (CED) software. Stimuli consisted of 10 repeats of 500 ms sound pulses followed by 500 ms silent periods. Frequency (11 pure tones ranging from 23 to 120 kHz) and SPL (9 levels ranging from 46 to 94 dB in 6 dB increments) were systematically altered for a total of 99 combinations. Individual action potentials from auditory afferents were too small to be individually identified and characterised amidst all the other neuronal activity in the nerve. Therefore, recordings for each train of 10 stimuli were root-mean-square transformed (time constant 0.66 ms) to convert the neuronal traces into positive displacements from zero and averaged. This allowed the neuronal response to sound to be characterised as an area, with units of µVs. An averaged response to each train of ten pulses and succeeding silent periods per sound intensity and frequency was produced in Spike2. The response area to 475 ms of sound stimulus (excluding the transient 'on' response immediately after the onset of a sound pulse) and an equivalent 475 ms in the succeeding silent period was measured in each averaged response. The mean areas of response (in microvolt s, µVs) during the presentation of each different sound stimulus was compared to the mean response during the subsequent silent period in each animal ($n$ = 5) using paired $t$-tests.

## Echolocation calling frequencies of co-occurring bats and insect call recordings

Echolocation calls of phyllostomid bats (Chiroptera: Phyllostomidae) native to Gorgona Island (*Gardnerycteris crenulatum*, *Tonatia saurophila* and *Micronycteris microtis*) were recorded in a small indoor flight cage (1.4 × 1.0 × 0.8 m) located in Gamboa, Panama, in which they were allowed to fly. The echolocation calls were recorded via an ultrasound condenser microphone (2–200 kHz frequency range, ± 3 dB frequency response between 25 and 140 kHz; CM16, CMPA preamplifier unit, Avisoft Bioacoustics, Glienicke, Germany) and real time ultrasound acquisition board (6 dB gain, 500 kHz sampling rate, 16 bit resolution; UltraSoundGate 116Hm, Avisoft Bioacoustics, Glienicke, Germany) connected to a laptop (Think Pad X220, Lenovo, Beijing, China), with a corresponding recording software (Avisoft RECORDER USGH, Avisoft Bioacoustics, Glienicke, Germany). The calls were analyzed with the sound analysis software Avisoft SASLabPro (5.2.15, Avisoft Bioacoustics, Glienicke, Germany), using automatic measurements. For details on the recordings and analysis, please refer to *Geipel et al., 2021*. Recording the bat echolocation calls followed the ABS/ASAB guidelines for ethical treatment of animals and were approved by the Government of Panamá (Ministerio de Ambiente permit SE/A-5–19) and the Smithsonian Tropical Research Institute (STRI ACUC protocol 2019-0302-2022).

Sound recordings of the male *C. gorgonensis* calling song were performed in a sound-attenuated booth at the Sensory Biology Lab, University of Lincoln at a temperature of 25 °C and relative humidity of 40%. The specimens were placed on a metallic screen cage at 10 cm from a 1/8" microphone (B&K Type 4138 omnidirectional microphone), connected to a 1/4" preamplifier (B&K Type 2670) and set to a conditioning amplifier (Nexus 2690-OS1). The microphone was calibrated at 94 dB SPL (re 20

µPa), using a B&K sound level calibrator (B&K Type 4231, Brüel & Kjaer, Nærum, Denmark). Data was obtained via an acquisition board (PCI-6110, National Instruments, Austin, TX, USA) and stored on a computer hard disk at a sampling rate of 512 kHz using the Polytec acquisition software (PSV 9.0.2, Polytec GmbH, Waldbronn, Germany). Sound was analyzed using Matlab (R2015a, The MathWorks, Inc, Natick, MA, USA) (*Figure 6—figure supplement 1C*).

## Statistical analyses

Using empirical data we tested the effect of cuticular pinnae on tympanal responses [in displacement amplitude (natural log transformed) and arrival time] to incident sound, we fitted linear mixed models (LMM) with angle (–10° to 10°, quadratic polynomial continuous variable) as a covariate and presence of pinnae (y/n), frequency (23, 40, and 60 kHz, categorical variable), tympanum (anterior or posterior) as fixed factors. We included the interactions between angle and pinnal presence and between pinnal presence and frequency. To model the curvature in the response surface of the pinnal enclosed tympanum, angle was fitted as a quadratic polynomial with 0° at point zero. The interaction of angle and pinnae was fitted as such to show the restriction of pinnal structures in both time and displacement to the response surface. To account for repeated measures of the same specimen, we nested leg (left or right) within individual specimens as a random factor. We carried out post hoc tests between pinnae (y/n) at each frequency using estimated marginal means from the package *emmeans* (*Lenth and Lenth, 2018*).

Using the same initial LMM model, we tested how sound pressure estimated from numerical models was related to angle (–10° to 10°, polynomial continuous variable), presence of pinnae (y/n), frequency (23, 40, and 60 kHz, categorical variable), tympanum (anterior or posterior) as fixed factors. Again, we include the interactions between angle and pinnae and between pinnae and frequency. We finally tested sound pressure based on 3D models with the presence of pinnae (y/n), frequency (23, 40, and 60 kHz, categorical variable), tympanum (anterior or posterior) as fixed factors, and with the inclusion of the interaction between pinnae and frequency. For both numerical and 3D models, we carried out post hoc tests between pinnae (y/n) at each frequency using estimated marginal means from the package *emmeans*.

Statistical tests and graphs were performed in R 4.0.0 (*R Development Core Team, 2021*) and all LMMs were run using the package lmerTest (*Kuznetsova et al., 2017*). Our data are freely available on the online Dryad repository (*Pulver et al., 2022*).

## Acknowledgements

This research is part of the project "The Insect Cochlea" funded by the European Research Council, Grant ERCCoG-2017–773067 to FM-Z; and was also funded by the Natural Environment Research Council (NERC), grant DEB-1937815 to FM-Z. We thank the Orthopterists' Society for aiding in funding the micro-computed tomography work of CW, for which some data has been used in this study, and to the University of Lincoln's School of Life & Environmental Sciences for CW's PhD studentship. The authors gratefully acknowledge Dr. Berthold Hedwig for sharing laboratory space, equipment and valuable advice. The authors thank the editors and reviewers for their comments and feedback in improving our manuscript.

## Additional information

### Funding

| Funder | Grant reference number | Author |
| --- | --- | --- |
| European Research Council | ERCCoG-2017-773067 | Fernando Montealegre-Z |
| Natural Environment Research Council | DEB-1937815 | Fernando Montealegre-Z |

The funders had no role in study design, data collection and interpretation, or the decision to submit the work for publication.

## Author contributions

Christian A Pulver, Formal analysis, Investigation, Visualization, Methodology, Writing – original draft, Performed acoustic experiments on live specimens, Designed and performed all 3D print model experiments, Constructed the speaker mount, automated the controls in the arena, collected and analysed data, Assisted with preparing the manuscript; Emine Celiker, Conceptualization, Software, Formal analysis, Validation, Investigation, Visualization, Methodology, Developed and performed all numerical models and simulations, Assisted with preparing the manuscript; Charlie Woodrow, Formal analysis, Validation, Investigation, Methodology, Designed and performed all 3D print model experiments, Illustrated all figures, conducted micro-computed tomography scans, Performed all post-image segmentation for numerical analyses and conducted 3D printing, Conducted all electrophysiological experiments; Inga Geipel, Formal analysis, Investigation, Methodology, Recorded insect and bat acoustic signals and provided bat call analysis, Assisted with preparing the manuscript; Carl D Soulsbury, Formal analysis, Supervision, Validation, Methodology, Writing – review and editing, Designed the statistical model and completed statistical analyses, Assisted with preparing the manuscript; Darron A Cullen, Investigation, Methodology, Validation, Visualization, Conducted all electrophysiological experiments, Performed and analysed neurophysiological audiograms, Edited the paper for eLife style; Stephen M Rogers, Formal analysis, Methodology, Performed and analysed neurophysiological audiograms, and produced the relevant figures; Daniel Veitch, Investigation, Assisted Christian Pulver with preliminary experiments; Fernando Montealegre-Z, Conceptualization, Resources, Supervision, Funding acquisition, Methodology, Project administration, Writing – review and editing, Led the lab, assisted with idea development, and provided equipment training. Performed acoustic experiments on live specimens, Conducted fieldwork and recorded insect and bat acoustic signals, Performed behavioural and neural audiograms. Illustrated and formatted figures, Conceived the idea of the project, Edited the paper for eLife style

## Author ORCIDs

Christian A Pulver ⓘ http://orcid.org/0000-0001-9197-9960
Emine Celiker ⓘ http://orcid.org/0000-0003-4988-7901
Charlie Woodrow ⓘ http://orcid.org/0000-0001-7342-0792
Carl D Soulsbury ⓘ http://orcid.org/0000-0001-8808-5210
Darron A Cullen ⓘ http://orcid.org/0000-0002-6287-5086
Daniel Veitch ⓘ http://orcid.org/0000-0003-4404-8498
Fernando Montealegre-Z ⓘ http://orcid.org/0000-0001-5186-2186

## Ethics

Experimental insect welfare during and post experiments follow procedures approved in Ethics Project "0245 The Insect Cochlea: directionality in the ear" at the University of Lincoln. Live experiment specimens and the colonies from which they are maintained are also covered by FMZ laboratory Ethics Project.

## Decision letter and Author response

Decision letter https://doi.org/10.7554/eLife.77628.sa1
Author response https://doi.org/10.7554/eLife.77628.sa2

---

# Additional files

## Supplementary files

• Transparent reporting form

## Data availability

Data files are available in Dryad (https://doi.org/10.5061/dryad.k0p2ngf8x).

The following dataset was generated:

| Author(s) | Year | Dataset title | Dataset URL | Database and Identifier |
|---|---|---|---|---|
| Montealegre-Z F | 2022 | Data from: Ear pinnae in a neotropical katydid (Orthoptera: Tettigoniidae) function as ultrasound guides for bat detection | https://dx.doi.org/10.5061/dryad.k0p2ngf8x | Dryad Digital Repository, 10.5061/dryad.k0p2ngf8x |

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
