## [Editor Report]

This study combines an impressive combination of experimental and computational approaches to probe the function of the cuticular pinnae, structures that form air-filled cavities around the tympanal ears on the forelegs of bush crickets. In many other species – including mammals – the external ears are known to play a critical role in helping to localize sounds. The results of this study show, however, that the very small resonant cavities formed by the pinnae in one particular bush cricket species are able to boost ultra-high frequency sound waves that lie well above the frequencies used for communicating with conspecifics. This raises the possibility that these structures may have evolved to assist bush crickets to detect the ultrasonic echolocation calls of their bat predators.

---

## [Decision Letter]

**Decision letter after peer review:**

Thank you for submitting your article "Ear pinnae in a neotropical katydid (Orthoptera: Tettigoniidae) function as ultrasound guides for bat detection" for consideration by *eLife*. Your article has been reviewed by 2 peer reviewers, and the evaluation has been overseen by a Reviewing Editor and Andrew King as the Senior Editor. The following individual involved in the review of your submission has agreed to reveal their identity: Heiner Romer (Reviewer #1).

Essential revisions:

1) Clarify and refocus the text: Both the introduction and discussion are hard to read for the *eLife* readership, which includes biologists, neuroscientists, and life scientists (etcetera) from all areas of biological research. The current manuscript is targeted at direct colleagues/specialists working on insect and bat hearing & coevolution and as a result, the text distracts from the main message that any reader should be able to distill and comprehend after reading. For example, the discussion covers 8 pages, which is too long compared to the length of the rest of the manuscript. Much of the discussion currently recaps many results that should be presented in the Results section. To improve the clarity and focus, we suggest you identify the key points that would matter for an *eLife* reader and cut back on non-essential details when their contribution to the main point is not critical for the scientific quality of the paper. You should consider concentrating on the main finding of the resonances within the space provided by the pinnae, and the resulting SPL boost at relevant ultrasonic frequencies, and how this supports your hypothesis that the pinnae may function as bat detectors.. Another aspect that has broader impact is the lack of evidence of their function for directional hearing. Other issues include that it is not clear to the reader if the presence of pinnae is a taxonomic trait characteristic of Copiphorini. If yes, it may be interesting to search for species that either live in a bat-free habitat (like Fullard´s studies on moths) or are diurnally active to contrast findings. One could hypothesize that in both cases the selection pressure for the evolution of pinnae to boost ultrasonic frequencies would not exist. All these and other related issues are solvable with a well-thought-out revision designed to serve the general *eLife* readership.

2) The introduction is a hard read for the many *eLife* readers who will not be familiar with tettigoniid ears due to much technical detail that distracts from the main message. Instead, the different anatomical parts of the system need to be explained clearly in the figures. These introductory figures need a major overhaul so anybody outside tettigoniid ear morphology can follow the introduction. Given the highly specialized hearing organ anatomical location, morphology and function it would help if the first figure places this unique hearing organ in its comparative context across other hearing organ morphologies and functions in a wider range of species. For example, currently, it takes too much effort to figure out that the pinnae are paired per ear and on the side. Please clarify this for a general *eLife* reader in Figure 1B. Further, many different terms seem to be used for the same structure: What is the spiracle, tympanal port, EC, slit, subslit cavity etc. This is also ablated in text /absent in figures.

3) Figures and supplements: The figure organization and presentation and supporting supplementary figures and information need to be streamlined and more focused. For example, Figure 5 represents three different and complementary approaches that would be logical to separate. Using these methods is an impressive feat, but we felt that a reader looking at the figure would struggle to come to the same conclusion due to the lack of clarity. Further, the large number of supplements can better be reduced and combined into actual supporting figures in the paper. For example, Figure 1 Supp1 can easily be panel C in Figure 1 (merge B-D in current Figure 1).

4) Figure 2

i) It includes a nice schematic cross-section of the ear, but this is needed in Figure 1 to follow the theory laid out in the introduction.

ii) Figure 2B, C: How do you explain the strong difference between experimental and numerical data for 60 kHz before and after pinnae ablation?

iii) Further, the symbols for pinnae intact or absent are not easy to distinguish in B, C.

iv) Finally, the high significance in B at 40 kHz seems improbable. Please double-check your outcomes for errors and correct them as needed. If there are no errors a clarification is in order so the reader can interpret the validity of this unusual outcome independently.

5) The scale in Figure 1 supp 1 is hard to read. To help the reader further, please indicate that the pinna is present in all species. The figure layout suggests the photos under the right animal have no pinnae.

6) Figure 3 sup1; the colors don't seem to match the axes which make this figure nonintuitive for our readership.

7) Figure 3 and 4: the axis in dB SPL should read dB, see also Figure 6. Please resolve this issue throughout the manuscript.

8) Figure 4 has 6 supplements; it will help the reader if the authors select the essential ones and incorporate those into the paper or one single strong supplement.

9) Figure 5:

i) Figure 5A, why is the tympanal response in the free sound field so strong for PTM, but relatively weak for ATM? Can the 13% larger pinnae volume explain such a difference?

ii) Figure 5C, lines 271ff and 339ff: This figure could be deleted, or replaced by a threshold curve, if available. It seems rather trivial that the amplitude of response increases with SPL. An easy-to-perform experiment is missing which would provide strong and convincing evidence for the function of pinnae: Recording afferent neural activity (or even single-cell activity of the T-fibre activity) before and after ablation of the pinnae. According to the biophysical experiments with the scaled printed ear model, we would expect a 20 – 30 dB increase in threshold after ablation. If this could be performed it would very much strengthen the manuscript.

iii) Figure 5B: the behavioural threshold at high frequencies between 90 and 120 kHz is remarkably low, apparently as a result of the pinnae. It would be worthwhile to discuss and compare this outcome with published data on other katydids and crickets.

iv) Figure 5C: the reader struggles to see the black and white symbols.

10) Figure 6 and Figure 6 suppl. 1: The spectrograms of the three bat species in Figure 6 are repeated in the suppl. Figure, which makes the supplement obsolete. Any essential information would be better included in Figure 6 instead.

11) The manuscript has many long sentences. As this makes the manuscript harder to read than it should be, we recommend that you try and avoid excessively long sentences (e.g. by using Macros for Word to automatically find and highlight sentences beyond a certain length). Further, the manuscript has a good number of acronyms, which do not serve our readership. Please replace non-standard acronyms with the actual name.

Line-by-line comments

L46: please fill out the sound pressure gains in dB.

L49: it would help the general reader to add the underlying reason, e.g. for enhanced detection of predatory echolocating bats.

L57: This seems to relate to the need to detect sound signals, both of conspecific mates and predators. The general reader would benefit from additional context.

L70ff: should the reader conclude that positioning the ears in the legs is a solution to the problem? How is this motivated, what is the hard evidence? Please fully substantiate/revise.

L77: Acoustic trachea is a confusing term.

L94: What is meant by phylogenetic context? Are pinnae a characteristic feature of Pseudophylline and Conocephaline katydids? Please make this point understandable for the general *eLife* reader.

L96: what is auditory morphology? Ear morphology?

L196-101: This section is unclear. A general reader can’t follow the logic of the dual-channel system consisting of the EC and spiracle. There is a tube (EC) and the spiracle seems to be its opening, so how should the reader interpret this as two channels? This section needs to be revised to serve the general *eLife* reader.

L103-106; this sentence is hard to follow. What is the main point?

L107: is this point about directional hearing? How should the general reader interpret this sentence and how can it be revised so it’s clear?

L117: please provide the approx. dB SPL of th’ low amplitude sound.

L136: It would be helpful to write how this contributes to sound detection and localization.

L140: The slits are first mentioned here, please explain to a general *eLife* reader.

L145 What is meant by guides? Here the lack of anatomical context and clear illustration is hindering the comprehension of the general reader. Where are the resonance cavities? Delineated by tympanum, spiracle and pinna? Please make this information more accessible for the general *eLife* reader.

L154-157: these details can go into the M and M.

L163 Define the concepts PTM and ATM and don't use acronyms in the text so general readers do not need to memorize specialist acronyms.

L164: this does not seem to be shown in Figure 2B, please resolve.

L168-170: Please explain in more detail so a general reader can follow.

L170: please substantiate and clarify why this finding demonstrates that resonance of the tympanum is the underlying reason.

L172: Doesn't the significance symbols from the post hoc analysis suggest high significance for 40 kHz?

L176: What is the N? Please resolve and clarify this throughout the manuscript, e.g. the entire Results section and in each figure caption in which data is presented.

L229-232: Please rephrase so the general *eLife* reader can comprehend.

L271-271: Some introductory sentences are needed to help the general reader comprehend. Please address that Figure 5 packs three very different methods into one, which may escape the attention of the general reader and in general requires further illustration / visual and text communication. Should the reader assume the neural audiogram uses spike density, how could the general reader easily grasp this? Please revise.

L280-281: Is this statement supported by statistics? Please show and clarify in the text.

L272-285: A weakness of this section is that it is unclear for a general reader what the basis is of the selected results that are presented and what the main points are. This needs to be resolved thoroughly to avoid the impression of cherry-picking.

L274: these points do not seem to be visible in Figure 5c.

L277: Is the resolvable signal statistically significant? Information that would enable the general reader to assess this independently is missing, please address fully.

L292: the reader is missing the statistical support for this section. Please fully resolve.

L298: please provide the missing information that the reader needs to understand why only gleaning bats are considered in this context.

Line 354: Please explain the discrepancy at 60 kHz or communicate the underlying limitation.

Line 1337: please clarify for the general *eLife* reader, are C-K all Copiphorine species?

---

## [Author Response]

Essential revisions:1) Clarify and refocus the text: Both the introduction and discussion are hard to read for the eLife readership, which includes biologists, neuroscientists, and life scientists (etcetera) from all areas of biological research. The current manuscript is targeted at direct colleagues/specialists working on insect and bat hearing & coevolution and as a result, the text distracts from the main message that any reader should be able to distill and comprehend after reading. For example, the discussion covers 8 pages, which is too long compared to the length of the rest of the manuscript. Much of the discussion currently recaps many results that should be presented in the Results section. To improve the clarity and focus, we suggest you identify the key points that would matter for an eLife reader and cut back on non-essential details when their contribution to the main point is not critical for the scientific quality of the paper. You should consider concentrating on the main finding of the resonances within the space provided by the pinnae, and the resulting SPL boost at relevant ultrasonic frequencies, and how this supports your hypothesis that the pinnae may function as bat detectors.. Another aspect that has broader impact is the lack of evidence of their function for directional hearing. Other issues include that it is not clear to the reader if the presence of pinnae is a taxonomic trait characteristic of Copiphorini. If yes, it may be interesting to search for species that either live in a bat-free habitat (like Fullard´s studies on moths) or are diurnally active to contrast findings. One could hypothesize that in both cases the selection pressure for the evolution of pinnae to boost ultrasonic frequencies would not exist. All these and other related issues are solvable with a well-thought-out revision designed to serve the general eLife readership.

We agree with the reviewers’ assessment and addressed these issues with several emendations and restructuring. Both introduction and discussion have been significantly shortened. For instance, the discussion was reduced from 8 manuscript pages to 5.5 pages. We now maintain the focus of the paper on the main role of the auditory pinnae in ultrasound capturing and prey detection, and have removed addition information and jargon that can distract the reader. We also produced a broad search of species with and without auditory pinnae and discuss potential function, including diurnal species or species not predated by bats. We have also explicitly defined objectives of the study and have restructured subsections of the results to match those of the *methods*, as suggested by the reviewers.

There are not of course some katydids existing devoid of bat predation, although the most remarkable studies focused on Pseudophyllinae, the conocephaloids have not been studied in such habitats. It begs an excellent question (perhaps copiphorines of southern Africa). We do provide arguments in the discussion describing different acoustic pinnal functions in conocephaloids affected by bats, but also discuss pinnae function of diurnal conocephaloids, which are mainly predated by birds, as a main point of an interesting topic that might generate interest and debate.

2) The introduction is a hard read for the many eLife readers who will not be familiar with tettigoniid ears due to much technical detail that distracts from the main message. Instead, the different anatomical parts of the system need to be explained clearly in the figures. These introductory figures need a major overhaul so anybody outside tettigoniid ear morphology can follow the introduction. Given the highly specialized hearing organ anatomical location, morphology and function it would help if the first figure places this unique hearing organ in its comparative context across other hearing organ morphologies and functions in a wider range of species. For example, currently, it takes too much effort to figure out that the pinnae are paired per ear and on the side. Please clarify this for a general eLife reader in Figure 1B. Further, many different terms seem to be used for the same structure: What is the spiracle, tympanal port, EC, slit, subslit cavity etc. This is also ablated in text /absent in figures.

We agree with the reviewers’ comments on the structure of the introduction and the use of acronyms. The Introduction has been improved and shortened to explain more substantially how the system functions using more context and improved phrasing. We have added illustrative panels and proper labels to Figure 1 to show the anatomy of the ear.

3) Figures and supplements: The figure organization and presentation and supporting supplementary figures and information need to be streamlined and more focused. For example, Figure 5 represents three different and complementary approaches that would be logical to separate. Using these methods is an impressive feat, but we felt that a reader looking at the figure would struggle to come to the same conclusion due to the lack of clarity. Further, the large number of supplements can better be reduced and combined into actual supporting figures in the paper. For example, Figure 1 Supp1 can easily be panel C in Figure 1 (merge B-D in current Figure 1).

The number of supplementary figures have been reduced, for example Figure 1 does not include supplements as its current version is self-explanatory. Figure 2 with two supplements, Figures 3-6 with one Supplement each. We also noticed that in the figures describing the experiments using 3D models and numerical models the results are based on the pinnal cavities and not on the tympanic membranes. The figures initially submitted refer to the tympanic membranes ATM (for anterior tympanic membrane) and PTM for anterior tympanic membrane, perhaps because we are used to these two acronyms in your research, but in reality we should have referred to the anterior and posterior pinnal cavities. Therefore, we have corrected this in the new version of the manuscript and figures.

4) Figure 2i) It includes a nice schematic cross-section of the ear, but this is needed in Figure 1 to follow the theory laid out in the introduction.

Thank you for pointing this out. We have included a cross section of the ear as an inset in panel B.

ii) Figure 2B, C: How do you explain the strong difference between experimental and numerical data for 60 kHz before and after pinnae ablation?

The experimental data shows mechanical responses of the tympanum while the numerical data is predicting sound pressure within the cavities. Thus, the mechanical responses are different (in magnitude and dynamics) than sound pressure. We argue that the tympanal membrane has a resonance close to the frequencies related to the calling song, that being 23 kHz. The experimental data show a mechanical response at 60 kHz with and without pinnae, albeit reduced and not different. On the other hand, the numerical data is measuring sound pressure inside the cavities. With the cavities intact (with pinnae), the sound pressure gain is much greater than with the pinnae ablated. The cavity induced pressure gains only work with pinnae, so the effect of a micro-cavity is quite pronounced even below their reported best resonance. At 110 – 120 kHz, the pinnae act like Helmholtz-like resonators able to capture and amplify diminishing ultra-high frequency sound waves in the reported range. The acoustic consequence of this effect imparts greater forces acting on the tympanal surface, and thus the mechanical response would be greater with pinnae than without. Of course, if we were able to use the input isolation platform with a loudspeaker able to produce frequencies tested in the numerical models. This is discussed in lines 342-352.

iii) Further, the symbols for pinnae intact or absent are not easy to distinguish in B, C.

Thank you for pointing this out. We corrected symbols for Figure 2B and C.

iv) Finally, the high significance in B at 40 kHz seems improbable. Please double-check your outcomes for errors and correct them as needed. If there are no errors a clarification is in order so the reader can interpret the validity of this unusual outcome independently.

This was an error. We apologise for uploading the wrong Figure 2B. The post hoc tests showed no significance at 40 kHz. The figure has been updated.

5) The scale in Figure 1 supp 1 is hard to read. To help the reader further, please indicate that the pinna is present in all species. The figure layout suggests the photos under the right animal have no pinnae.

We have edited Figure 1, and removed supplementary figure (in response to one of the comments of another reviewer). The new Figure 1 is presented in higher resolution and shows four panels. Panel D was used to indicate variation in pinnae across katydid species.

6) Figure 3 sup1; the colors don't seem to match the axes which make this figure nonintuitive for our readership.

We have produced a better version of the figures with axis colours matching the respective spectra: grey for calling songs, and blue for the resonances of pinnae cavities.

7) Figure 3 and 4: the axis in dB SPL should read dB, see also Figure 6. Please resolve this issue throughout the manuscript.

This has been edited.

8) Figure 4 has 6 supplements; it will help the reader if the authors select the essential ones and incorporate those into the paper or one single strong supplement.

We agree that Figure 4 has too many supplements. We now offer 1 relevant supplement with 2 panels (formerly 4 and 6).

9) Figure 5:i) Figure 5A, why is the tympanal response in the free sound field so strong for PTM, but relatively weak for ATM? Can the 13% larger pinnae volume explain such a difference?

The PTM does exhibit a broader and greater response at high frequencies; higher than our limited 60 kHz (lines 168-171; see also Jonsson et al. 2016). We don't know if these differences related to the area on each tympanic membrane available for positioning the laser bean and obtain a single point data, the angle of the laser beam, or differences in membrane sensitivity. We briefly discuss potential differences of the tympanal membranes (lines 412-424), as we don't have evidence of the discrepancies created by comparing two tympanal regions with different mechanics, as mentioned above.

ii) Figure 5C, lines 271ff and 339ff: This figure could be deleted, or replaced by a threshold curve, if available. It seems rather trivial that the amplitude of response increases with SPL. An easy-to-perform experiment is missing which would provide strong and convincing evidence for the function of pinnae: Recording afferent neural activity (or even single-cell activity of the T-fibre activity) before and after ablation of the pinnae. According to the biophysical experiments with the scaled printed ear model, we would expect a 20 – 30 dB increase in threshold after ablation. If this could be performed it would very much strengthen the manuscript.

This was a whole nerve recording in which the weakest responses of auditory receptors could not be distinguished from the ambient neuronal activity. Therefore, doing a threshold is problematic and our approach was to determine when the response became significantly different from the background, which is not the same thing. Figure 5C shows how responsiveness changes with frequency, the rate at which responsiveness increases with SPL, and where these responses saturate. So even though it would be expected that responsiveness increases with SPL, this figure allows different frequencies to be compared. We have replaced the panel with a response profile to 76 dB SPL. Unfortunately, we do not have animals to perform the suggested experiments as our colonies basically disappeared during the pandemic, and it was too difficult to remove pinnae from animals after the recording prep had been set up, so we could not take a comparative ‘before and after’ approach. This will form the basis of future experiments, when our insect colonies are repopulated.

iii) Figure 5B: the behavioral threshold at high frequencies between 90 and 120 kHz is remarkably low, apparently as a result of the pinnae. It would be worthwhile to discuss and compare this outcome with published data on other katydids and crickets.

We agree with the reviewers’ suggestion. We have inserted in lines 279-285: “Audiograms showed that the startle response of females decline sharply for stimuli between 20 kHz and 35 kHz, however, response increases at around 35 kHz, and remains essentially constant at higher frequencies over the entire tested frequency range (Figure 5B; Table 3). A decline in threshold was found at the resonances of the tympanal pinnal cavities (90 kHz to 120 kHz) 59.28 ± 1.80 dB SPL (Figure 5B).

And added to the discussion: on lines 483-488: This demonstrated acute sensitivity (or predator escape response) to frequencies matching both the tympanal pinnal cavities and the call of echolocating bats. The low behavioural threshold at high frequencies between 90 and 120 kHz, has been reported for other species. The average startle behavioural threshold in *C. gorgonensis* was 59.28 ± 1.80 dB SPL (Figure 5B), which is comparable to the behavioural response in *Neoconocephalus ensiger* (Faure and Hoy, 2000).”

iv) Figure 5C: the reader struggles to see the black and white symbols.

This panel was replaced with a response profile curve.

10) Figure 6 and Figure 6 suppl. 1: The spectrograms of the three bat species in Figure 6 are repeated in the suppl. Figure, which makes the supplement obsolete. Any essential information would be better included in Figure 6 instead.

The new version of this supplementary figure includes only spectrograms of other katydids not included in the main research of the paper, but that share same habitat with *Copiphora gorgonensis*. For bat calls we included the identified bats we show in the main figure, but also other recorded unidentified bats.

11) The manuscript has many long sentences. As this makes the manuscript harder to read than it should be, we recommend that you try and avoid excessively long sentences (e.g. by using Macros for Word to automatically find and highlight sentences beyond a certain length). Further, the manuscript has a good number of acronyms, which do not serve our readership. Please replace non-standard acronyms with the actual name.

We are thankful for the reviewers’ comments and suggestions. To facilitate ease of reading and interpretation of results, we have clarified and rewrote relevant segments of the paper to address the long sentences. Acronyms have been removed. We also chose more general language to describe mechanistic functions and systems.

Line-by-line commentsL46: please fill out the sound pressure gains in dB.

We have inserted (20–30 dB), line 47.

L49: it would help the general reader to add the underlying reason, e.g. for enhanced detection of predatory echolocating bats.

We have added “ enhanced detection of predatory echolocating bats” line 52.

L57: This seems to relate to the need to detect sound signals, both of conspecific mates and predators. The general reader would benefit from additional context.

We have edited this phrase accordingly (lines 58-59).

L70ff: should the reader conclude that positioning the ears in the legs is a solution to the problem? How is this motivated, what is the hard evidence? Please fully substantiate/revise.

We thank the reviewers with this comment. We have clarified this in the introduction (line 70-76).

L77: Acoustic trachea is a confusing term.

We agree and removed “also called the acoustic trachea.”

L94: What is meant by phylogenetic context? Are pinnae a characteristic feature of Pseudophylline and Conocephaline katydids? Please make this point understandable for the general eLife reader.

We edited the paragraph (line 96-115) as suggested by the reviewers. Now, we show how common pinnae are across the phylogeny by providing percentage of pinnal bearing katydids based on our approximations from the online database (The Orthoptera Species Files). And we expand on this in the discussion.

L96: what is auditory morphology? Ear morphology?

Clarified in the new version of this paragraph (line 98).

L196-101: This section is unclear. A general reader can’t follow the logic of the dual-channel system consisting of the EC and spiracle. There is a tube (EC) and the spiracle seems to be its opening, so how should the reader interpret this as two channels? This section needs to be revised to serve the general eLife reader.

We appreciate this comment, and we have rewritten the segment with the new version of the paragraph, and removed the unnecessary details that included findings of other authors. This allows us to tackle the problem directly: the function of the auditory pinnae was unknown (lines 96-107).

L107: is this point about directional hearing? How should the general reader interpret this sentence and how can it be revised so it’s clear?

We agree with the reviewers concerns about clarity. To clarify the potential of directional hearing with external inputs, we inserted line 112-115 “… produced the strongest responses when stimuli was presented directly opposite the cavity entrances, and weakest contralaterally to the same stimuli. This difference in intensity between the two ears potentially contributes to directional orientation in rainforest katydids.”

L117: please provide the approx. dB SPL of the low amplitude sound.

in our efforts to reduce the length of the introduction, we removed this sentence and replaced it with a more general review. But used the topic in the discussion to justify that even small displacement of the tympanic membranes will trigger an auditory response (L368-380).

L136: It would be helpful to write how this contributes to sound detection and localization.

We appreciate the suggestion, and inserted (line 111-115) to answer this. “It was reported that diffraction of very short wavelengths along the tympanal cavity entrances (or slits, Figure 1B) produced the strongest responses when stimuli was presented directly opposite the cavity entrances, and weakest contralaterally to the same stimuli. This difference in intensity between the two ears potentially contributes to directional orientation in rainforest katydids.”

L140: The slits are first mentioned here, please explain to a general eLife reader.

We agree that this terminology requires more explanation. We edited “cavity entrances (or slits, Figure 1B)” Line 112.

L145 What is meant by guides? Here the lack of anatomical context and clear illustration is hindering the comprehension of the general reader. Where are the resonance cavities? Delineated by tympanum, spiracle and pinna? Please make this information more accessible for the general eLife reader.

We thank the reviewers for this comment. We have recorded this last sentence in the introduction and removed the word ‘guides’ to make it clear that we test the hypothesis that pinnae work as ultrasound detectors (line 131). We have also changed Figure 1A to include a panel to illustrate the anatomical features of the ear.

L154-157: these details can go into the M&M.

We have not removed this sentence as we feel it provides an introductory sentence to the Results section where we remind readers of the approaches employed. Lines 146-152.

L163 Define the concepts PTM and ATM and don't use acronyms in the text so general readers do not need to memorize specialist acronyms.

We have removed acronyms. Only use the in some of the figure labels to optimise space and explain in the caption.

L164: this does not seem to be shown in Figure 2B, please resolve.

We apologise for this error. It has been corrected.

L168-170: Please explain in more detail so a general reader can follow.

We are thankful for this comment. We have elaborated a bit more using the following sentence: “This demonstrates that pinnae do not enhance auditory perception of the carrier frequency in *C. gorgonensis*, and that the observed displacement, event after ablation, results from the fact that tympanal natural resonance produces maximum vibrational amplitude at 23 kHz, the carrier frequency of the species call”. Lines 159-163.

L170: please substantiate and clarify why this finding demonstrates that resonance of the tympanum is the underlying reason.

The new sentence in previous comment covers this issue.

L172: Doesn't the significance symbols from the post hoc analysis suggest high significance for 40 kHz?

We apologise for this error. It has been fixed in figure 2.

L176: What is the N? Please resolve and clarify this throughout the manuscript, e.g. the entire Results section and in each figure caption in which data is presented.

We have included “*n* = 7;” to each reported mean; lines 167-175.

L229-232: Please rephrase so the general eLife reader can comprehend.

Thank you for pointing this out. We removed “see next section” “an equivalent” and inserted “a V shaped cavity”; lines 223-227.

L271-271: Some introductory sentences are needed to help the general reader comprehend. Please address that Figure 5 packs three very different methods into one, which may escape the attention of the general reader and in general requires further illustration / visual and text communication. Should the reader assume the neural audiogram uses spike density, how could the general reader easily grasp this? Please revise.

We’ve added the following sentence (lines 277-286):

Extracellular whole auditory nerve recordings, made with suction electrodes, were used to produce neural audiograms (Figure 5C). The auditory nerve is a mixed nerve, containing the axons of many neurons beside those of auditory afferents, leading to high levels of activity unrelated to auditory stimuli. Furthermore, the high firing rates and small amplitudes of auditory afferent action potentials spread across a population of responsive afferents meant that individual action potentials could not be resolved (Figure 5—figure supplement 1, A). Instead, the sum neuronal activity in the auditory nerve during sound stimuli was compared with that during silent intervals. Responsiveness was measured by root-mean-square transforming the data (time constant = 0.66 ms) and measuring the area under the curve (Figure 5—figure supplement 1, A; red, during sound stimulation, blue in between sound stimuli).

L280-281: Is this statement supported by statistics? Please show and clarify in the text.

We stated that “Amplitudes of response increased almost linearly with sound pressure...” We could not perform a formal test to show that this was linear given the small sample size, so we have removed the words ‘almost linearly” and rephrase the entire paragraph (Lines 294-304).

L272-285: A weakness of this section is that it is unclear for a general reader what the basis is of the selected results that are presented and what the main points are. This needs to be resolved thoroughly to avoid the impression of cherry-picking.

We have adjusted this paragraph slightly to make our main point clear, which is that the greatest response at every SPL was at the calling song frequency (Lines 310-324).

L274: these points do not seem to be visible in Figure 5c.

We do not entirely understand this comment, because we can clearly see our white points on the coloured mesh (the ‘sound on’ response), denoting a significant difference from the background response at that combination of frequency and intensity. We have replaced this panel with response profile curve as suggested by one of the reviewers, and moved the colour plot as supplementary of Figure 5. Nevertheless, we have increased the size of the white and black points resting on the mesh to hopefully make them clearer.

L277: Is the resolvable signal statistically significant? Information that would enable the general reader to assess this independently is missing, please address fully.

We outlined the statistical approach in our *Methods* but not here, which we have now rectified by inserting “(P<0.05, paired t-test) “ into Figure 5 caption. We have also uploaded a table of original data and statistical tests to Dryad, which support our assignations of significance in Figure 5C.

L292: the reader is missing the statistical support for this section. Please fully resolve.

We cite Figure 5 supplement and rectified the stats procedure in lines 923-935 of the *methods*.

L298: please provide the missing information that the reader needs to understand why only gleaning bats are considered in this context.

This is an outstanding question. From previous studies on bats and their associated guilds of Central America, evidence from roosts of gleaning bats showed katydids as a staple of their diet. Many neotropical gleaning bats can eavesdrop on the acoustic signals of katydids, this is presumably why many neotropical katydids show low calling rates. It has also been shown that although many gleaning bats can detect motionless prey on leaves, they seem to prefer moving katydids. So although the bats exemplified here are mostly known as gleaning species, they can eavesdrop on singing katydids as *M. microtis*. We have mentioned this in the discussion (lines 465-467) to provide that fine knowledge to readers.

Line 354: Please explain the discrepancy at 60 kHz or communicate the underlying limitation.

We have mentioned this in the *methods* (line 608-610). The sole reason is hardware, the probe of the speaker produces all sorts of distortions above 60 kHz.

Line 1337: please clarify for the general eLife reader, are C-K all Copiphorine species?

Figure 1 was changed completely, and we indicate in the caption if this figure the specific names and subfamily of each pinnae example (Lines 1303-1319).